

# Computing the eigenstate localisation length
# at very low energies from Localisation Landscape Theory

Sophie S. Shamailov[1⋆], Dylan J. Brown[1,2],
Thomas A. Haase[1] and Maarten D. Hoogerland[1]

**1** Dodd-Walls Centre for Photonic and Quantum Technologies, Department of Physics,
University of Auckland, Private Bag 92019, Auckland 1142, New Zealand.
**2** Present Address: Light-Matter Interactions for Quantum Technologies Unit,
Okinawa Institute of Science and Technology Graduate University,
Onna, Okinawa 904-0495, Japan.

⋆ sophie.s.s@hotmail.com

## Abstract

While Anderson localisation is largely well-understood, its description has traditionally been rather cumbersome. A recently-developed theory – Localisation Landscape Theory (LLT) – has unparalleled strengths and advantages, both computational and conceptual, over alternative methods. To begin with, we demonstrate that the localisation length cannot be conveniently computed starting directly from the exact eigenstates, thus motivating the need for the LLT approach. Then, we confirm that the Hamiltonian with the effective potential of LLT has very similar low energy eigenstates to that with the physical potential, justifying the crucial role the effective potential plays in our new method. We proceed to use LLT to calculate the localisation length for very low-energy, maximally localised eigenstates, as defined by the length-scale of exponential decay of the eigenstates, (manually) testing our findings against exact diagonalisation. We then describe several mechanisms by which the eigenstates spread out at higher energies where the tunnelling-in-the-effective-potential picture breaks down, and explicitly demonstrate that our method is no longer applicable in this regime. We place our computational scheme in context by explaining the connection to the more general problem of multidimensional tunnelling and discussing the approximations involved. Our method of calculating the localisation length can be applied to (nearly) arbitrary disordered, continuous potentials at very low energies.

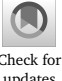
# 1 Introduction

Anderson localisation [1,2] is a universal wave interference phenomenon, whereby transport (i.e. wave propagation) is suppressed in a disordered medium due to dephasing upon many scattering events from randomly-positioned obstacles. This can be understood from Feynman's interpretation of quantum mechanics, where one must sum over all possible paths from the initial to the final points of interest to obtain the total transmission probability. The random positions of the scatterers guarantee dephasing between the different paths, leading to an attenuation of the amplitude of the wavefunction. First discovered in the context of quantised electron conduction and spin diffusion [3], Anderson localisation of particles thus provides direct evidence for the quantum-mechanical nature of the universe at a small scale.

In particular, Anderson localisation is characterised by an exponential decay in the tails of the wavefunction with a length scale known as the localisation length [3]. The computation of this key variable is not straight-forward. For continuous systems, a rough estimate can be obtained by setting the renormalised diffusion coefficient, derived in the limit of weak scattering where it is only slightly reduced from its classical value, to zero [2, 4, 5]. While the resulting analytical formula is not expected to be accurate, it is of course convenient, and is thus used by many researchers [6–9]. The diffusive picture is in general often employed to describe Anderson localisation, even though it is strictly inapplicable in this limit [7,9]. A rigorous calculation can be performed using Green's functions [2,4,5,10], but it requires many assumptions regarding the nature of the disorder and is quite involved. On the other hand, Green's functions can be used to extend the classical diffusive picture into the weakly-localised regime by computing the correction to the diffusion coefficient [2, 4, 5, 7], and even push this picture into the strongly localised limit by making the renormalised diffusion integral equation self-consistent [2, 4, 5, 7, 11].

Another approach to obtain the localisation length is the Born approximation, commonly utilised for weak scattering [6,8,12]: here, one takes the total wave in the extended scattering body as the incident wave only, assuming that the scattered wave is negligibly small in comparison. Understandably, this method is inaccurate for strong disorder. Exact time-dependent simulations with the Schrödinger [6, 8, 13, 14] or Gross-Pitaevskii [15, 16] equations can be used instead, but this approach is very time-consuming and yields little insight into the physics. Finally, access to the localisation length directly through the eigenstates of the Hamiltonian is

hampered by practical considerations (as we shall show below).

Other, more model-specific methods have also been employed in the literature: [17] solved the Schrödinger equation via a random walk on a hyperboloid, [18] derived a non-linear wave equation to extract the Lyapunov exponents corresponding to the linear problem of interest, [19] solved the kicked-rotor model analytically, and [20] derived analytical expressions relevant for the weak disorder limit.

For discrete models, a plethora of methods to calculate the localisation length likewise exists. The most renowned is of course the transfer matrix method, allowing for the calculation of Lyapunov exponents and thus the localisation length [21–29]. Such calculations have commonly been used to confirm the predictions of finite scaling theory [24, 27]. While often used together, transfer matrices and Lyapunov exponents have been combined with other elements to obtain the localisation length: the former with analytical continuation [30] to compute moments of resistance and the density of states, and the latter in a perturbative expansion, with numerical simulations of a quantum walker [31]. The Kubo-Greenwood formalism has also proved highly successful [24, 32, 33].

Green's functions have been as invaluable for discrete systems as for continuous [10, 12, 13, 24, 34, 35], allowing for renormalisation techniques to be applied [35, 36], or alternatively scattering matrices, treated with the Dyson equation [10]. Out of these references, [34] examined the off-diagonal elements of the Green's matrix as a localisation order parameter, [13] the distribution of eigenstates which was related to the spatial extent of the eigenstates, [10] the characteristic determinant related to the poles of the Green's function, and Ref. [35] developed a renormalised perturbation expansion for the self energy. Recursion formulae encoding the exact solution [37, 38] can also sometimes allow one to calculate the localisation length (and the density of states [38]).

Out of the studies above, one-dimensional (1D) [6, 13, 17–19, 21–23, 30, 31, 34, 35, 37, 38] and two-dimensional (2D) [6–8, 10, 12, 14, 15, 20–28, 36] models have been numerically explored far more thoroughly than three-dimensional (3D) [6, 27, 34], simply because of the increased computational requirements of higher-dimensional spaces. Possibly the most heavily studied model of localisation is the Anderson model, also known as a tight-binding Hamiltonian [4, 10, 12, 13, 22–27, 29, 30, 32, 34–37, 39–44], but other examples include the kicked rotor [19] (formally equivalent to the Anderson model), the Lloyd model [13, 21], the Peierls chain [38], a quantum walker [31], and the continuous Schrödinger equation [13, 14, 17], with either a speckle potential [7, 16], delta-function point scatterers [6, 10], or more realistic Gaussian scatterers [8, 15].

It is worth noting that for 2D continuous potentials with arbitrary disorder, there is no numerically-exact or even a fairly accurate, approximate method to compute the localisation length, leaving the direct integration of the time-dependent Schrödinger equation as the only currently viable approach.

Meantime, a break-through new theory – coined Localisation Landscape Theory (LLT) [45–51] – was developed recently, completely revolutionising the field. It allows for intuitive and transparent new insights into the physics, as well as a practical, efficient way of performing calculations. To give a brief overview, this theory relies on the construction of a function, the localisation landscape, which governs all the low-energy, localised physics. One can treat finite problems so that boundary effects are accounted for, and yet push the algorithms to very large system sizes, where alternative methods are completely impractical. The validity of this theory is not restricted to a specific noise type, making it widely applicable to a range of problems. An effective potential can be constructed, such that quantum interference effects can be captured instead by quantum tunnelling through this effective potential (but this is restricted to low energies, as we shall show). One can predict the main regions of existence (referred to as "domains") of the low-energy localised eigenstates, reconstruct the eigenstates on these

domains, as well as compute the associated energy eigenvalues. Thus, Anderson localisation can be fully reinterpreted in this picture, including the energy dependence of the localisation length (so far, qualitatively). Very recently, LLT has been used to support an experimental study of Anderson localisation [52].

In this paper, we search for a way to calculate the localisation length for an arbitrary disordered, continuous potential at very low energies. We begin by showing that the localisation length cannot be efficiently extracted from the eigenstates of the Hamiltonian. From there, we turn to LLT, and confirm that the Hamiltonian with the effective potential has very similar low-energy eigenstates to that with the physical potential, which justifies basing our calculation of the localisation length on a semiclassical treatment of tunnelling in the effective potential. Indeed, we demonstrate how the localisation length at very low energies can be obtained from LLT, a method that can be applied to continuous systems with any everywhere-positive potential (a required condition for LLT to apply), for any strength of the disorder, at very low energy where the eigenstates are strongly localised. Our description is in 2D, a 1D version is much simpler and can be implemented with no additional effort, while a 3D version can be eventually developed by a direct analogy.

Thus, we extend LLT by developing a method for the computation of the localisation length at very low energies in 2D systems. The main achievement lies in finding an efficient way of evaluating the exponential decay cost for crossing domain walls – barriers in the effective potential of LLT separating neighbouring domains. We discuss how our method fits in to the extensive literature on multidimensional tunnelling, and then test it against the results of exact diagonalisation. We also describe the mechanisms by which the eigenstates extend to cover larger areas at higher energies, and explain why our method cannot capture this behaviour, which can no longer be viewed as a simple tunnelling process in the effective potential.

The paper is structured as follows. We begin by introducing the system of interest in section 2, and proceed to demonstrate what can and cannot be learned from an exact diagonalisation of the Hamiltonian in section 3. In section 4, we show that the effective potential of LLT can be used to access the exponential decay in the low-energy eigenstates of the physical potential by comparing the eigenstates of the Hamiltonian with the two potentials. Then, in section 5, we extend known LLT to calculate the localisation length at very low energies, as defined by the length scale of exponential decay in the tails of the eigenstates of the Hamiltonian, and directly test the method by comparison to exact eigenstates. This method breaks down at higher energies, together with the tunnelling picture, as we describe in detail in section 6. In the course of our work, we develop a simple and practical approximation to multidimensional tunnelling, discussed in section 7, which has many potential applications in other contexts. Conclusions are presented in section 8 and several ideas are discussed as directions for a possible forthcoming investigation.

## 2 System of interest

We consider a (non-interacting) particle of mass $m$ confined to a 2D plane, whose motion is restricted to a rectangular region defined by $x \in [0, L]$ and $y \in [0, W]$. At the boundaries of this rectangular region, we impose Dirichlet boundary conditions, requiring the wavefunction to vanish. The particle moves in an external potential $V(x, y)$, so that the Hamiltonian is simply

$$H = -\frac{\hbar^2}{2m} \nabla^2 + V(x, y). \tag{1}$$

Because we are interested in studying Anderson localisation, the potential $V(x, y)$ is taken as a sum of $N_s$ randomly-placed Gaussian peaks of the form

$$V_0 \exp\left\{-\frac{(x-x_0)^2 + (y-y_0)^2}{2\sigma^2}\right\}, \tag{2}$$

constituting what is known as "point-like" disorder, chosen for its lower percolation threshold [8].

This system could be experimentally realised with cold atoms as in [15], where an attractive 2D trap is used to contain atoms in a planar geometry, a repulsive custom potential generated by a spatial light modulator (SLM) allows the atoms to be confined to, for example, a rectangular box, and Gaussian point-like scatterers are generated by imaging squares of light produced by the SLM.

Next, we must introduce a set of dimensionless units, to be used throughout the paper. Let $\ell$ be a typical physical length scale relevant for the problem (for example, $\ell \sim \sigma$). Lengths will be measured in units of $\ell$, energy is units of $E_0 = \hbar^2/(2m\ell^2)$, and time in $t_0 = \hbar/E_0$. Typically, for a cold-atom experiment such as [15], $\ell \sim 1\ \mu$m, $E_0 \sim 1$ nK $\times k_B$, and $t_0 \sim 5$ ms.

Note that the coordinates $(x_0, y_0)$ of the Gaussian scatterers are drawn from a uniform distribution of *half-integers* between $[0, L/\ell]$ and $[0, W/\ell]$, respectively. In all the simulations to follow, $L/\ell$ and $W/\ell$ are further chosen as integers. This restriction is imposed to stay in line with the discrete nature of the pixelated SLM used in [15] to both set the geometry and produce the scatterers. In the case of this experiment, one could reasonably choose $\ell$ to be the length of the side of the squares imaged on the SLM to produce the disorder.

The density of the scatterers is a more meaningful quantity to quote than their number, especially when one wishes to examine the effect of system size. Therefore, we define a dimensionless density, referred to as the fill factor, $f$, as

$$f = \frac{N_s \ell^2}{LW}. \tag{3}$$

Later in the article, we will discuss time-dependent simulations (direct integration of the Schrödinger equation) as a benchmark for a comparison to our LLT calculations. Such simulations will be performed in the transmission scenario, where we add empty "reservoirs" on either side of the disorder where the potential is zero. These occupy $x \in [-R, 0]$, $y \in [0, W]$ (first reservoir, $R_1$) and $x \in [L, L+R]$, $y \in [0, W]$ (second reservoir, $R_2$). Usually, we will choose $R = 30\ell$, just large enough to contain the initial condition that will be used. In the transmissive scenario, a wavefunction with centre-of-mass translation starts out in $R_1$ and goes through the disorder, finally arriving in $R_2$. The Dirichlet boundary conditions are applied at the boundaries of this extended rectangular region, including the reservoirs.

The initial condition we will use in this set up is a 1D Gaussian wavepacket[1] (Gaussian along $x$ and uniform along $y$), which is fairly wide in position space and therefore has a rather localised energy distribution. The functional form is simply

$$\psi = \exp(ik_0 x)\exp\left[-\frac{(x+R/2)^2}{4\bar{\sigma}^2}\right], \tag{4}$$

where we leave out the normalisation constant. Commonly, we will choose $\bar{\sigma} = 5\ell$, and use either of $k_0 = 1/\ell$ or $k_0 = 0.5/\ell$.

Time-dependent partial differential equations (PDEs) are solved using Matlab's modern PDE solver, `solvepde.m`, with Dirichlet boundary conditions.

---

[1] The use of similar probing waves was independently suggested by [6] and used in the experiment [53].

# 3 Exact diagonalisation

Our overall aim is to predict the localisation length for the system in section 2. Since the system size is finite, the potential is continuous, and does not have the required statistics for the Green's functions method to be helpful, the standard techniques cannot help us. The only available (accurate) methods are time-dependent Schrödinger simulations or exact diagonalisation. The former requires running individual simulations at each energy and involves some ambiguity, arising from what exactly is done and how – such analysis is (mostly) left for a separate paper [54, 55]. In the present section, we pursue the latter, since all the information is certainly contained in the Hamiltonian, its eigenstates, and its spectrum. We therefore begin our investigation by directly diagonalising the Hamiltonian and inspecting the eigenstates and energies, with the goals of (a) gaining intuition for our system and (b) checking whether useful quantitative predictions may be readily obtained in this framework. For this purpose, we adapt the new algorithm developed in [56] (implemented in the "Chebfun" toolbox for Matlab [57] for 1D operators) by extending it to 2D. In general, our results support the well-known fact that the localisation length increases with energy. We find that the localised eigenstates lie at low energies, and the degree of localisation decreases as the energy increases. This can be easily seen by eye when inspecting the eigenstates, plotting their amplitude, $|\psi|$. An example is shown in Fig. 1, depicting nine low-energy eigenstates for a particular noise realisation. Overall, as energy increases, the weight of the eigenstates spreads out over a larger area (see Fig. 3 of [45] for another example). This process, however, is not monotonic: occasionally we encounter very localised states with a fairly high energy, where most of the energy comes from the rapidly changing wavefunction rather than the spatial extent and the associated potential energy. Also quite intuitively, if $f$ or $V_0$ are increased, the strength of localisation increases and the area within which the weight of the eigenstates is contained shrinks. Figure 2 demon-

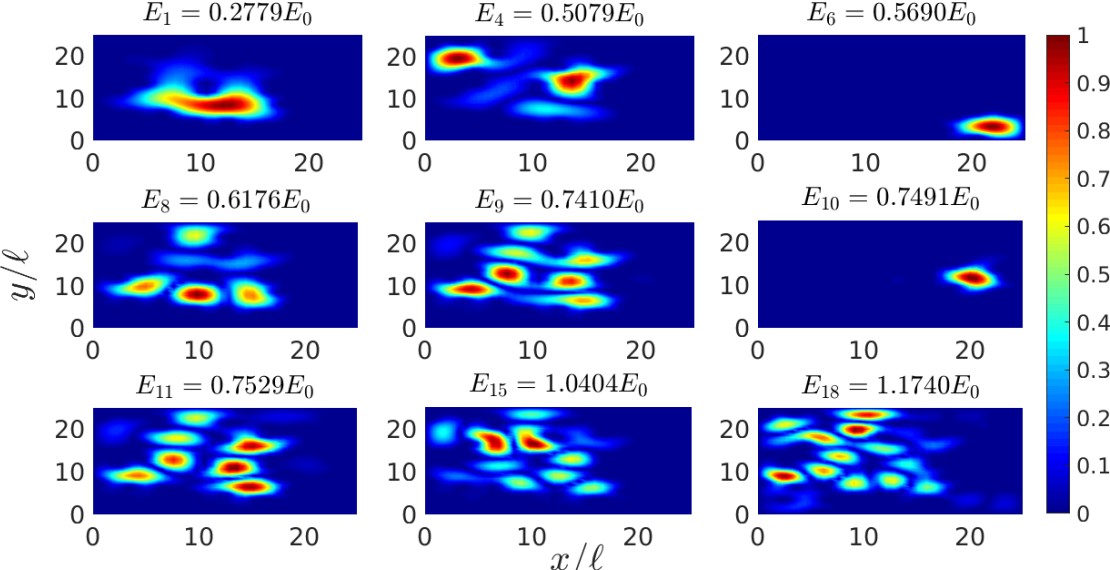

Figure 1: Nine low-energy eigenstates of the Hamiltonian for a given noise realisation with $L = W = 25\ell$, $f = 0.1$, $V_0 = 20E_0$, $\sigma = \ell/2$, showing the absolute value of the eigenstates as a colour-map. Note that all eigenstates are normalised such that the maximum is one so that the values can be read on the same colour bar. We see that overall, the spatial extent of the eigenstates increases with energy, quoted above each panel. However, occasionally, very localised states are encountered at higher energies, on account of the considerable kinetic energy such eigenstates carry.

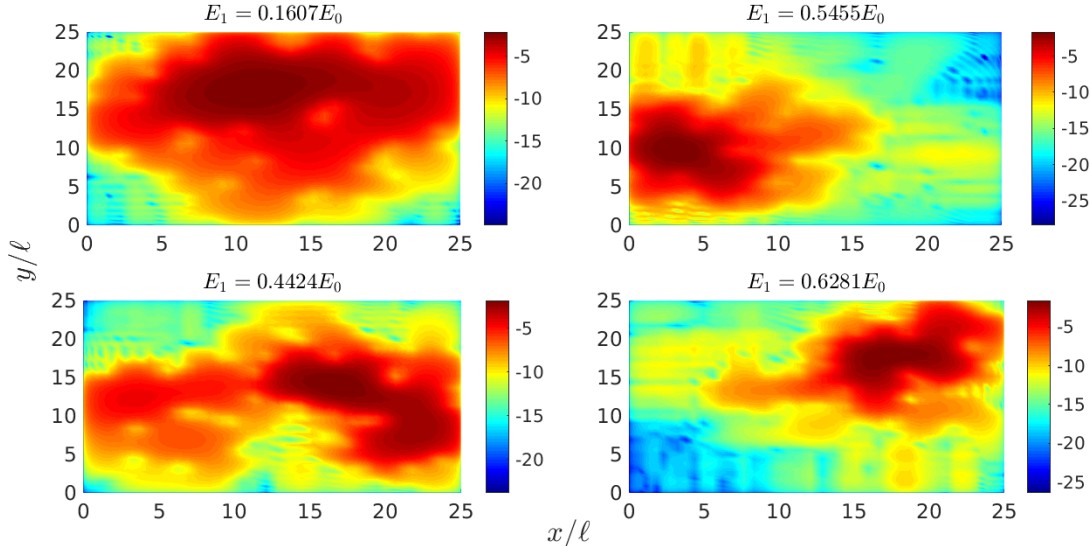

Figure 2: The lowest eigenstate of the Hamiltonian for some noise realisations with $L = W = 25\ell$ and $\sigma = \ell/2$, showing the logarithm of the absolute value of the eigenstates as a colour-map. Top left: $f = 0.1$, $V_0 = 10E_0$, top right: $f = 0.2$, $V_0 = 10E_0$, bottom left: $f = 0.1$, $V_0 = 20E_0$, bottom right: $f = 0.2$, $V_0 = 20E_0$. We observe that the degree of localisation is controlled both by the density of the scatterers and their height. The energy eigenvalue is quoted above each panel: it increases as the area of the (node-free) localised mode decreases. Note that the horizontal and vertical stripes seen most prominently in the right-hand-side panels are an artefact of the diagonalisation algorithm (at the given spatial resolution used) and are non-physical.

strates this by visually comparing the lowest energy eigenvector for different combinations of $f$ and $V_0$ (different noise realisations are used for each panel). We see that both the fill factor and the scatterer height are equally important parameters, influencing localisation properties just as strongly.

Increasing the width of the scatterers $\sigma$ also leads to stronger localisation (not illustrated), because the area occupied by the Gaussian peaks increases, but the dependence on the scatterer width is not methodically explored here. Note, however, that when the width of the scatterers becomes sufficiently large, there is a decrease in the randomness of the potential as we approach the limit where the entire system is covered by overlapping potential bumps (the same of course happens as the fill factor is increased strongly). Once this regime is reached, localisation weakens with further increases of the fill-factor and the scatterer width.

The shape of the scatterers also naturally plays a role, but as long as the ("volume") integral over a single scatterer is kept constant, the specific functional form is expected to have a much weaker effect on the physics than $f$ and $V_0$. The shape of the scatterers influences the spectral properties of the disordered potential, the relation of which to a (possible) mobility edge could be investigated in the future.

Next, let us consider how the localisation length may be extracted from the exact eigenstates of the Hamiltonian. By definition, the localisation length is the length scale on which the localised states decay exponentially, far away from the region where their main weight is concentrated. This decay can be seen in Fig. 2 as a change of colour from dark red to red to orange to yellow to green to blue, as the wavefunction gradually drops by orders of magnitude. The localisation length increases with energy, depends on the strength of the disorder,

and should only be discussed in a configuration-averaged context.

If we inspect any one given eigenstate, assuming the energy is sufficiently low or locali­sation is strong enough, there is usually only one peak – one local maximum – in $|\psi|$. If we temporarily place our origin there and vary the azimuthal angle $\theta$, then the curve $|\psi(r)|$ along different directions will certainly be different depending on $\theta$. Still, we could average these curves over $\theta$, and attempt fitting an exponential function to the tail of the resultant. If the peak is located in a corner of our rectangular system, for example, the average should only be taken over those angles along which one has reasonable extent along $r$.

However, as energy increases (or localisation decreases due to changes in parameters), the eigenstates develop a multi-peak structure: there are several "bumps" (see Fig. 1), and it is not clear where to place our origin. Furthermore, the energy eigenvalues are of course quan­tised, so any extracted localisation lengths from single-peak eigenstates need to be averaged over noise realisations, only using eigenstates of roughly the same energy (binning within a reasonable range). This makes such an approach very limited.

Now, a very common solution to this problem – heavily used in the literature (e.g. [14, 16, 58–62]) – is to compute the spatial variance of the localised states instead. Since we are working in 2D, we could tentatively examine the quantity

$$\left[\Delta x^2 \Delta y^2\right]^{1/4}, \tag{5}$$

where the variance along $x$ is

$$\Delta x^2 = \left\langle x^2 \right\rangle - \left\langle x \right\rangle^2 = \int_0^L dx \int_0^W dy \, x^2 \, |\psi|^2 - \left[\int_0^L dx \int_0^W dy \, x \, |\psi|^2\right]^2, \tag{6}$$

assuming the wavefunction is normalised to one, and $\Delta y^2$ is defined similarly.

Figure 3 shows a typical low energy eigenstate, plotting $|\psi|$ on a linear scale. The small-amplitude yet large-scale structure seen on the logarithmic plots of Fig. 2, capturing the ex­ponential decay of the eigenstates away from their main region of existence, is completely invisible on such a plot. When there is a single main "bump" in the eigenstates, the variance-based length scale of (6) reports mostly on the width of the main peak (seen in Fig. 3, for example) – analogous to the full-width-at-half-maximum or the standard deviation of a Gaus­sian peak. It measures the size of the main bump, and carries only indirect information on the exponential decay in the tails. In cases when there are smaller, secondary bumps in the eigen­states, their presence increases the variance, even if their width and decay rate are identical

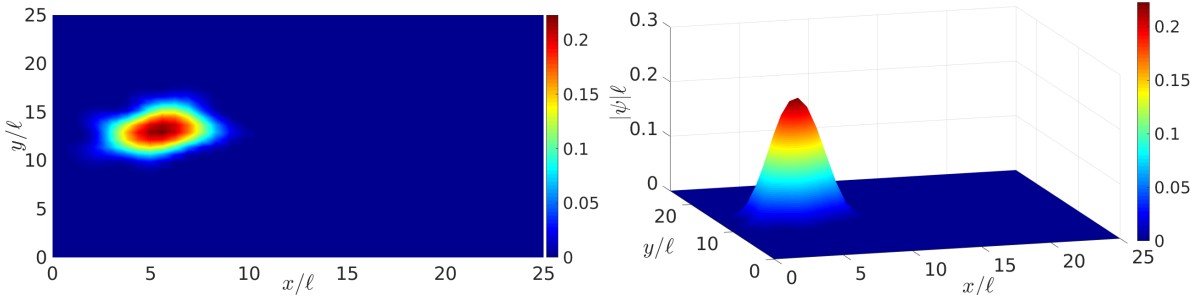

Figure 3: The lowest eigenstate of the Hamiltonian for a given noise realisation with $L = W = 25\ell$, $f = 0.2$, $V_0 = 20E_0$, $\sigma = \ell/2$, plotting $\ell\,|\psi|$ as a colour-map. The exponential decay away from the main region of existence of the eigenstate is unresolvable on a linear scale.

to those of the main bump. Therefore, the variance does not report on the localisation length, as such. We thus advise caution when using the variance to quantify localisation properties, a common practice in the literature.

# 4 The effective potential

Since exact diagonalisation cannot help us to efficiently extract the localisation length, we turn to LLT for a solution. The method we develop heavily relies on the effective potential introduced in this theory, and in this section, we justify its applicability to the problem at hand.

The key object of LLT is the localisation landscape $u$, defined by the PDE $Hu = 1$, where $H$ is the Hamiltonian [45]. The associated effective potential $W_E$ is simply given by $W_E = 1/u$. So far, LLT has produced several extremely useful results involving $W_E$ which allow to make physical predictions for a system with real potential $V$ – in our case, a disordered one. In particular, $W_E$ controls the regions of localisation of the eigenstates at different energies, the density of states according to Weyl's law, and the decay of the eigenstates through the valley lines of $u$ (paths of steepest descent starting at the saddle points of $u$ and ending at the minima, or terminating by exiting the system) according to the Agmon distance [48]. While the authors of [48, 49] motivate this remarkable success of the effective potential by an auxiliary wave equation, it appears that $W_E$ may, to a good approximation, be able to replace $V$ in the real Schrödinger equation, directly in the Hamiltonian, simply based on its successful use in place of $V$ in so many different formulae.

Ultimately, the main advantage of using the effective potential for us will lie in applying a semiclassical approximation to describe tunnelling at low energies in this landscape, but the semiclassical theory is an approximation to the full quantum-mechanical problem, and so before we explore the additional complexity of this approximation, we should check whether the substitution is valid in the full quantum mechanical treatment. This can be achieved by comparing the eigenstates in the two potentials and checking for similarity, which will justify the application of semiclassical tunnelling theory based on the effective potential $W_E$ to predict the behaviour of the eigenstates in the physical potential $V$.

In order to solve the stationary PDE for $u$ in large systems, it is best to use the domain decomposition method, an implementation of which is available using the legacy solver of the PDE toolbox in Matlab [63].

Before we begin, one may wonder whether the low-energy, localised states seen in Figs. 1 and 2 are simply trapped in local minima of the potential $V$, formed by surrounding Gaussian scatterers. When examined, the effective potential $W_E$ resembles the physical potential $V$ quite closely, as demonstrated in Fig. 4. The scatterer positions in $V$ largely coincide with peak positions in $W_E$, but the latter is a smoothed-out version of the former (on a length-scale which depends on $V(x, y)$), as discussed in [48]. In particular, while $V$ has clear gaps between scatterers (as long as the fill factor and scatterer width are not too great to cause significant scatterer overlap), $W_E$ has continuous potential ridges that encircle domains, allowing for classical trapping in these regions (this was also pointed out in [48]). Meantime, due to the smoothing, $W_E$ has lower peaks than $V$ (which is more noticeable at weaker disorder), and an almost constant, non-zero background value away from these peaks.

We note that since $W_E$ inherits so many of its features from $V$, it is also intrinsically a random potential, and will give rise to Anderson localisation (as was already realised in [48]). These quantum interference effects in $W_E$ will be similar to those in $V$ in as much as the two potentials are similar, but of course there will be differences in the localisation properties as well: for example, the lower peaks in $W_E$ would cause weaker localisation than one would

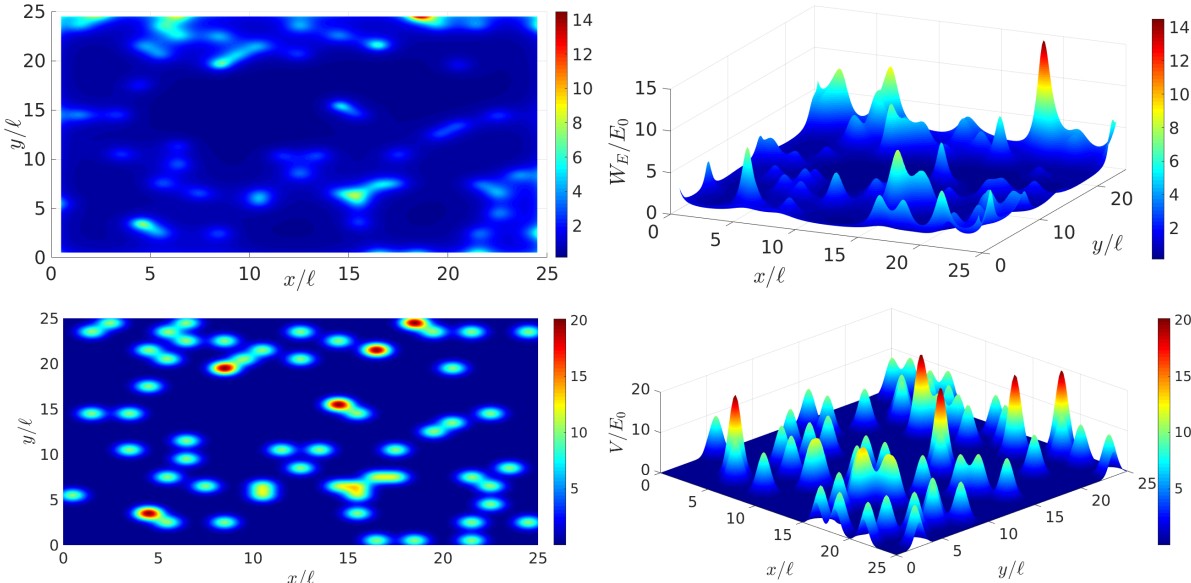

Figure 4: The effective potential $W_E$ (top panels) and the physical potential $V$ (bottom panels) with $L = W = 25\ell$, $f = 0.1$, $V_0 = 10E_0$, $\sigma = \ell/2$, viewed from the top and from the side. The peak ranges of $W_E$ correspond to the valley lines of $u$ and govern both the localisation regions of the eigenstates and their decay outside of their main domains of existence. While the potential barriers of $W_E$ are located largely at the positions of the scatterers in $V$, $W_E$ can be thought of as a smoothed out version of $V$, so that the clear gaps between the scatterers in $V$ are annealed, leading to the formation of proper local wells that can support classical trapping. The smoothing operation also has the effect of creating lower peaks in $W_E$ compared to $V$, which is relatively more significant for weak disorder, at the expense of creating an almost-constant, non-zero background value to the potential away from the peaks. In addition, $W_E$ inherits the random nature of $V$, and is capable of supporting Anderson localisation.

have in $V$.

Now, according to LLT, the valley lines of $u$ – collectively referred to as the "valley network" – divide the system into "domains" [45] (see the top panel of Fig. 6 for an illustrative example). The valley lines of $u$ are of course the peak ranges of the effective potential, simply due to the inverse relationship between $u$ and $W_E$. Therefore, the domains are surrounded by potential barriers and constitute the regions of localisation of low-energy eigenstates. Looking at Fig. 1, we point out that the various bumps visible in the eigenstates are forced to zero at the valley lines of the localisation landscape $u$, so that the weight of the eigenstates (at low energies) rests within the domains of LLT. Increasing the energy of the eigenstates enables the wavefunction to cross some of the potential barriers separating the domains and spread out further [48] (a detailed discussion of this phenomenon at higher energies is given in section 6).

We now proceed to check whether the eigen-states and -energies of $H$ with $W_E$ are similar to those of $H$ with $V$. To some extent, this is indeed the case, as demonstrated in Fig. 5. The energy spectrum seems very similar up to a global energy shift, first proven to exist and derived in [48], while the eigenstates themselves are closely correlated for sufficiently low energies. We find that for eigenstates that are localised to a handful of domains, involving fundamental local modes (i.e. there is only one density peak per domain), the similarity between eigenstates obtained using $V$ and $W_E$ is immediately obvious. Once localisation is weakened (e.g. by increasing the energy) to allow the occupation of many domains (possibly in excited local

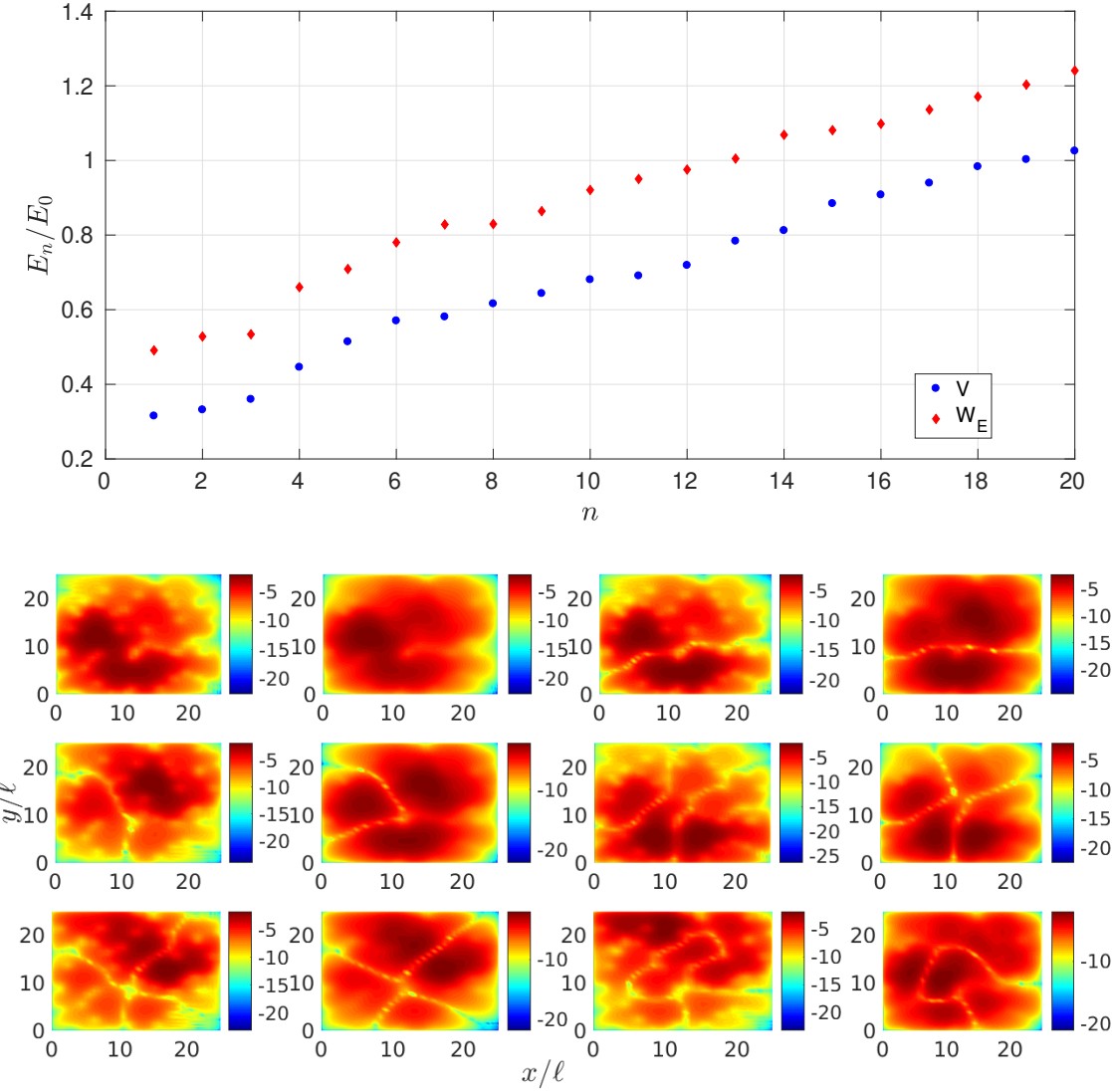

Figure 5: Low-energy eigenspectrum (top) and six of the lowest eigenstates with $L = W = 25\ell$, $f = 0.1$, $V_0 = 10E_0$, $\sigma = \ell/2$, showing the logarithm of the absolute value of the eigenstates as a colour-map (bottom). A direct comparison is drawn between the spectrum of the Hamiltonian with potential $V$ and with $W_E$ for the same noise realisation. The eigenvalues seem very similar, up to a global energy shift. In the bottom panel, going across the rows, we plot consecutively the $n^{\text{th}}$ eigenstate using $V$ and the $n^{\text{th}}$ eigenstate using $W_E$, alternating between the potentials before increasing $n$. Thus the first and second panels can be directly compared, the third and fourth, etc. Up to the fifth eigenstate, the correlation between the mode shapes is clear. From the sixth eigenstate onward, there is no visible relation between the eigenmodes of the Hamiltonian with the two potentials.

states), the correlation is lost. If Anderson localisation is strengthened (by increasing either or all of $V_0$, $f$, $\sigma$), more low-energy eigenstates match between the spectra of $H$ with $V$ and $H$ with $W_E$, and the agreement between the eigenstates is improved. We will discuss this further in section 6.

As a final note, if one evolves the same initial wavepacket in $V$ compared to $W_E$, one finds that transmission in the effective potential always happens more readily than in the real. This may be explained by the observation that the eigenstates of $H$ with $W_E$ are somewhat more extended than the exact and have higher overlaps. Moreover, we would expect Anderson localisation in $W_E$ to be weaker due to the lower peaks, which would lead to the same effect.

To conclude, we have shown that the lowest energy eigenstates in $V$ are similar to those in $W_E$. This will later allow us to apply a semiclassical approximation to tunnelling in $W_E$ and use it to make quantitative predictions about the decay of eigenstates in $V$, thus granting access to the localisation length.

# 5 Eigenstate localisation length

In this section we extend LLT to compute the localisation length for very low energy, maximally localised eigenstates, defined as the length scale of exponential decay in the tails of the eigenstates of the Hamiltonian. A combination of several LLT concepts allows for the development of a general methodology that can be applied to other systems, with other kinds of disorder, or in other dimensions.

In the regime where our calculation is applicable, we explicitly test our ideas by direct comparison to exact eigenstates and find good agreement. We highlight the unavailability of other reliable methods for the purpose of comparison and validation of our new technique. For example, the transfer matrix method is commonly used for discrete systems, and may be extended to 1D continuous systems [64]), but to the best of our knowledge, not to 2D. Previous papers that have used point-like disorder have faced a similar problem: Refs. [8,15] ran time-dependent simulations to extract the localisation length from the density decay rate, but were not able to compare their results to any other accurate or reliable computation.

In principle, we could compare our LLT calculation to time-dependent simulations, using a translating Gaussian wavepacket – an excellent choice of an initial condition to detect localisation. In practice, in order to have a sufficient energy range over which the LLT results are valid so as to accommodate the Gaussian in this narrow interval, localisation must be very strong indeed. In this regime, edge effects (described in more detail later) become important and cause the localisation lengths obtained from LLT and time-evolution to differ. Time-dependent simulations can, however, be used outside of the regime of applicability of the LLT calculation and be qualitatively tested for consistency with the indication provided by exact eigenstates regarding the question "in which way does the LLT calculation fail at higher energies, and how does it deviate from the true result?".

## 5.1 Outline of the LLT method

Recall that LLT has taught us that the low-energy eigenstates are localised inside domains of the valley network [45], and must tunnel through the peaks of the effective potential in order to spread to neighbouring domains (this is in contrast to the physical potential $V$, where there are gaps between scatterers, with the domains connected classically[2]). Within any given domain, there is nothing to induce exponential decay – the decay does not happen continuously (as

---

[2]This statement holds at reasonable fill-factors and scatterer widths. If either parameter is increased excessively such that the scatterers join and form closed regions in the plane, then classical trapping becomes possible.

commonly believed), but in discrete steps, every time the wavefunction crosses a valley line [48]. Furthermore, valley lines which are not part of a closed domain (referred to as "open" valley lines below) are irrelevant, as the wavefunction simply goes around them without losing amplitude.

Because of its prime importance to this section, we repeat here (from the original LLT papers [48,49]) the definition of the energy-dependent quantity known as the Agmon distance, which controls the decay of the eigenstates outside of their main domain of existence:

$$\rho_E(\mathbf{x_0}, \mathbf{x}) = \min_{\gamma} \left( \int_{\gamma} \Re \sqrt{2m[W_E(\mathbf{x}) - E]}/\hbar \; ds \right). \tag{7}$$

Since only the real part of the square root is used, the integrand is zero if $E$ exceeds $W_E$ at position $\mathbf{x}$. The integral should be minimised over all possible paths $\gamma$ going from $\mathbf{x_0}$ to $\mathbf{x}$, and $ds$ is the differential arc length. If we have an eigenstate peaked at position $\mathbf{x_0}$ inside some given domain, then it will have amplitude at position $\mathbf{x}$ outside of this main domain bounded by

$$|\psi(\mathbf{x})| \lesssim |\psi(\mathbf{x_0})| \exp\left[-\rho_E(\mathbf{x_0}, \mathbf{x})\right]. \tag{8}$$

As the authors of [48] point out, the formula (7) is commonly encountered in the context of the Wentzel–Kramers–Brillouin (WKB) approximation in 1D (and higher dimensions), and constitutes a semiclassical approximation of multidimensional tunnelling. The inequality (8) assumes that the connection between the wavefunction at points $\mathbf{x_0}$ and $\mathbf{x}$ is quantum mechanical tunnelling through the potential barriers between them. In 1D, Ref. [48] has shown that (8) can be used to predict the shape of the eigenstates very closely.

If we approximate the domains on average as circular in shape and denote the diameter $D$, then every distance $D$, the wavefunction undergoes a decay. The cost of crossing a valley line will be bounded below by the Agmon distance $\rho_E$, such that the amplitude of the wavefunction drops by at least a factor of $\exp(-\rho_E)$ on average every time. If we assume for the moment that $\rho_E$ faithfully captures the decay rate, combining these two quantities, we see that the localisation length is simply given by

$$\xi_E = D/\rho_E, \tag{9}$$

where the subscript $E$ on $\xi$ stands for "eigenstate". Remarkably, the difference between $D$ and $\xi_E$ was already realised in [35].

Now, evaluating $\rho_E$ between any two arbitrary points in the $x - y$ plane is extremely difficult, as discussed in section 7. However, this is not strictly necessary for our purposes. With the understanding that the system is divided into network domains, with every closed domain containing a unique maximum of $u$, we can estimate the Agmon distance between the minima of $W_E$ (equivalently, the maxima of $u$), considering only nearest neighbour domains. In other words, if we have two neighbouring domains (which share some common segment of domain walls), we aim to find the least-cost path, according to the Agmon measure, that connects the two unique maxima of $u$ which reside in these domains. Evaluating $\rho_E$ along this path would then be straight-forward.

Again, formally, finding the true least-cost path is a difficult task. We have found an approximate solution to this problem that seems much simpler to implement compared to all currently known alternatives, while not sacrificing much in terms of accuracy at all (see section 7 to gain perspective). Recall the valley lines are the paths of steepest descent, starting from each saddle point and ending at minima of $u$ (valley lines may also terminate by exiting the system). Consider now curves that start from the saddle points and follow paths of steepest *ascent*, ending at maxima of $u$. Each saddle point thus links two maxima of $u$, and the curve

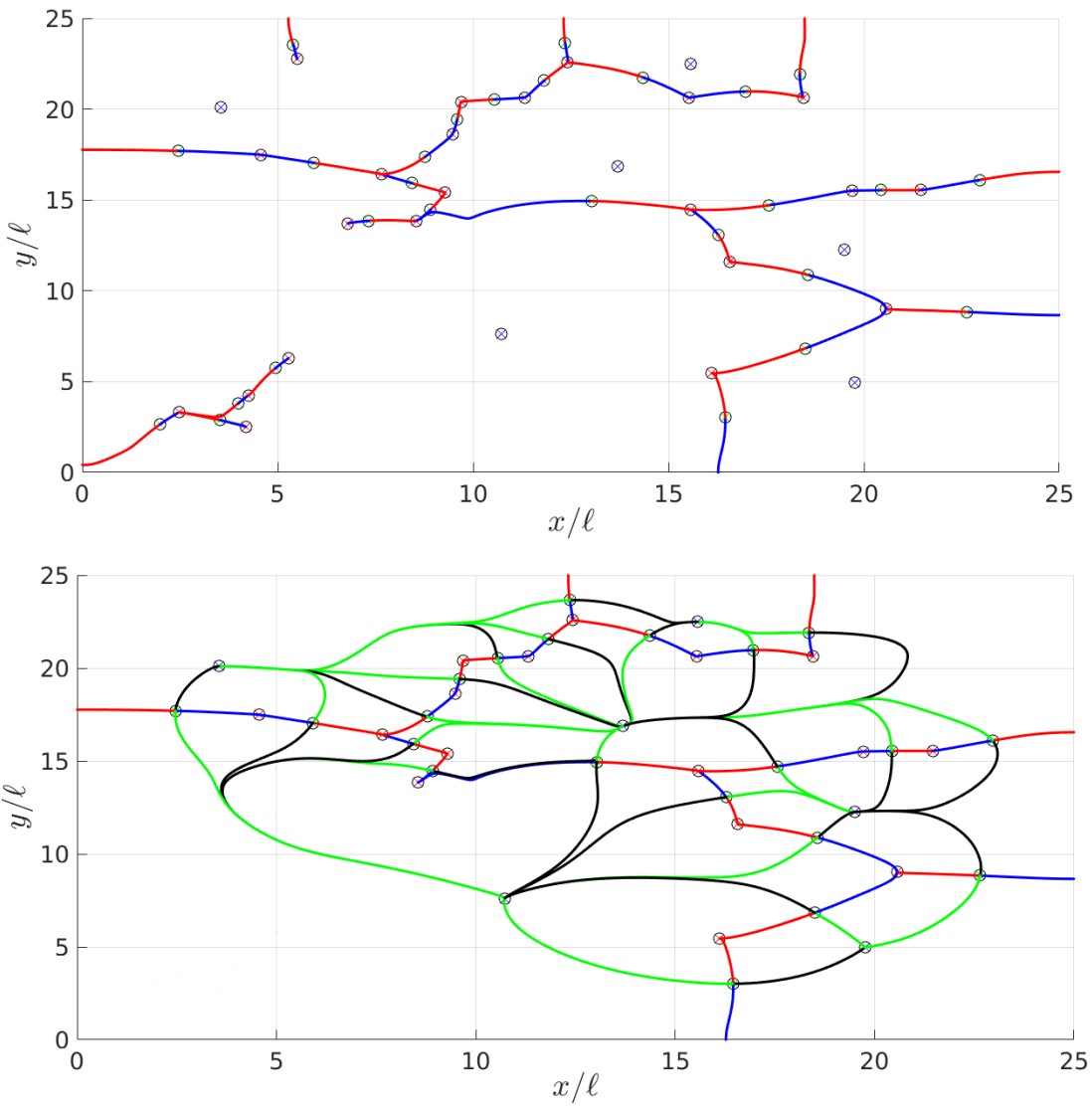

Figure 6: The original valley network (top) for some given noise realisation with $L = W = 25\ell$, $f = 0.06$, $V_0 = 5E_0$, $\sigma = \ell/2$, and the same network after all "open" valley lines have been removed (bottom). Both panels plot the valley lines in red and blue (different colours are used simply to make it easier to see the structure of the network). The extrema of $u$ are also shown as symbols (maxima in blue, minima in red, saddles in green). The bottom panel displays in addition all candidate approximate paths of least cost with respect to the Agmon metric as green and black lines (again, for more accessible visual interpretation), connecting neighbouring maxima of $u$ through the linking saddle points.

formed in this way is the lowest-lying path on the inverse landscape $W_E$ that connects the two minima of $W_E$ in question. Figure 6 first shows an example of the valley network as originally defined [45], and then with open valley lines removed (as they do not matter for eigenstate confinement and decay) and the candidate minimal paths connecting maxima of $u$ through the saddle points overlaid.

We will use these paths to compute $\rho_E$ between any two neighbouring maxima of $u$. First of all, we highlight that the Agmon distance is an energy-dependent quantity. Thus, along each path, the integral must be done separately at each energy of interest, $E$. Now, generally speaking, any two neighbouring domains have several common saddles on the shared section of their domain walls (see Fig. 6 for an example). At each energy, we must choose the minimal path which has the smallest Agmon integral out of the finite, discrete number of available options (which is computationally trivial). The path integral along that curve then becomes the Agmon distance $\rho_E$ between the domain maxima in question at the energy considered. This must be done for all neighbouring domains and at all energies in any given landscape $u$.

One may wonder, at this point, how well does our approximation capture the "real" Agmon distance, obtained by proper path minimisation, as described in section 7. We have tested this

Table 1: A comparison of the Agmon distance found using the approximate LLT path (the path of minimal cost out of the finite set of lowest-lying paths on $W_E$ that connect neighbouring minima of $W_E$ through the saddle points), to the real semiclassical result, obtained by solving differential equations, as described in section 7. In this case, we used a single noise realisation with $L = W = 25\ell$, $f = 0.06$, $V_0 = 21.33E_0$, $\sigma = 0.48\ell$, and three domain pairs, the results for which are separated by horizontal lines in the table. For each domain pair, we computed the Agmon distance at several energies, until the domains became classically connected and the cost vanished. In the one case where the table entry is missing (replaced by a hyphen), the true semiclassical path could not be found. It is immediately clear that the LLT approximation of the Agmon distance is a very close one, and that in all cases, our method only slightly overestimates the true minimal cost. This is a small price to pay for the remarkable computational advantages of our scheme compared to the real semiclassical solution (see section 7).

| $E/E_0$ | LLT approx. | True semiclassical |
|---|---|---|
| 0 | 6.8349 | 6.2586 |
| 0.1 | 5.7521 | 5.3130 |
| 0.2 | 4.3178 | 4.0233 |
| 0.3 | 2.0687 | 1.9918 |
| 0.4 | 0.9004 | 0.8831 |
| 0.5 | 0.2545 | 0.2491 |
| 0.6 | 0 | 0 |
| 0 | 6.6619 | 6.1395 |
| 0.1 | 5.0021 | 4.6693 |
| 0.2 | 2.4690 | 2.3319 |
| 0.3 | 0 | 0 |
| 0 | 5.4327 | 5.1905 |
| 0.1 | 4.0763 | 3.9064 |
| 0.2 | 1.9386 | 1.8750 |
| 0.3 | 0.2674 | — |
| 0.4 | 0 | 0 |

for several examples by solving the semiclassical equations and comparing the Agmon integral to that taken over the minimal lowest-lying path on the surface of $W_E$. We found that the true minimal path always lies very close to the minimal lowest-lying path and the integral along the latter is only slightly greater than the smallest possible cost obtained by proper path minimisation; the results are summarised in Table 1.

In the decay picture painted so far, restricting our consideration exclusively to neighbouring domains does not introduce an additional level of approximation: we only need to know the average cost of crossing from one domain into another, and decay over large distances can be simply composed of several such domain-to-domain tunnelling events. That is, our calculation only requires the computation of *local* quantities, which makes it largely system-size independent. Indeed, up to finite size effects which change the spacing of the valley lines at small system sizes (as studied in [55,65]), averaging over a few large systems will give the localisation length to the same precision as averaging over a bigger number of smaller systems: the only important factor is how many typical domains (for the area) and domain-pairs (for the tunnelling coefficient) are averaged over, not whether they are in one or several valley networks.

The next question is whether the Agmon distance $\rho_E$ faithfully captures the decay rate between neighbouring domains: after all, it is a lower bound on the decay coefficient, not an estimate thereof. We test this in the bottom panel of Fig. 7 (see the next subsection for details), finding that the Agmon distance itself systematically underestimates the true decay rate seen in the exact eigenstates. Therefore, rather than choosing the minimal-integral path, we take the average of the path integrals over *all* candidate paths from LLT (lowest-lying paths going through the saddle points), to obtain what we will coin the "mean" Agmon distance, $\bar{\rho}_E$. As we demonstrate in the top panel of Fig. 7 below, this method of computation actually captures the true decay rate much better, so we proceed with the understanding that

$$\xi_E = D/\bar{\rho}_E. \tag{10}$$

Note that this modified decay rate fully obeys the Agmon inequality (8), and that this is an advancement of semiclassical multidimensional tunnelling, as so far, it has only been possible to calculate the lower bound of the decay rate, but not an approximation of the real value (see section 7 for a further discussion).

As pointed out, $\bar{\rho}_E$ between neighbouring domains is an intrinsically energy-dependent quantity. Once the energy is so high that the saddle points of the candidate paths on the effective potential $W_E$ are below $E$, the cost of crossing from one domain to the other vanishes: $\bar{\rho}_E$ becomes zero as breaks develop in the domain wall separating the two maxima of $u$ (valley lines only effectively constrain eigenstates if $u < 1/E$, evaluated on the valley lines [45]). For our computation of $\xi_E$, we need the average of all non-zero $\bar{\rho}_E$ across the 2D system as a function of energy, but we also need to compute the domain area to extract the diameter, $D$. This requires integrating over the individual domain areas (at $E = 0$), averaging over all domains, assuming the area is that of a circle, and computing the diameter. However, as energy goes up and domain walls break down, domains effectively *merge*, so that the area increases with energy as well. Thus, in our calculation, domains are merged once $\bar{\rho}_E$ between them vanishes.

To summarise, the main steps of the calculation are as follows. Take a precomputed valley network, remove any open valley lines and calculate all the "candidate minimal paths" connecting saddles to maxima of $u$. Next, identify the valley lines (and potentially segments of the system boundary) that form the domain walls for each domain and perform local, on-domain integrals (e.g. to find the domain area, in which case the integrand is one). From here, identify all saddles linking any two neighbouring domains, calculate the path integral of the Agmon distance over all linking paths between them, and finally obtain $\bar{\rho}_E$ by averaging

over these integrals (including any paths that give a vanishing cost) at every energy. Then, for each noise configuration, the mean of $\bar{\rho}_E$ is computed over all neighbouring domain pairs, and the mean domain area yields the diameter $D$. Both of these quantities are energy dependent: zero-cost links are excluded from the average of $\bar{\rho}_E$ and domain areas are merged as the walls between them break down. Finally, many noise configurations need to be averaged over to get a reasonable estimate of the localisation length.

Note that an analogue of our LLT calculation cannot be usefully performed by using $V$ directly, instead of the effective potential $W_E$. This is because the exponential cost of crossing most domain walls would be zero (exceptions would be caused by scatterer overlap), as the scatterers in $V$ are separated by gaps. In other words, since classical trapping in $V$ is not possible (at reasonable fill factors and scatterer widths), a semiclassical tunnelling picture would predict no exponential decay. This is in addition to the fact that in order to find the domains, one needs the localisation landscape $u$ ($1/V$ would not yield closed domains in the valley network due to gaps between the scatterers). Thus, LLT is essential for our method and one could not avoid using it.

We remark that this calculation can be performed for any given localisation landscape as long as it has (appropriate) extrema. This includes, in particular, cases when the potential $V$ is regular and Anderson localisation is impossible. The resulting "localisation length" is then of course meaningless. It is up to the researcher performing the calculation to identify cases when one is dealing with localisation before attaching any significance to the result. This can be done by examining the fundamental on-domain eigen-energies, and ensuring that they are randomised, as explained in detail in [45, 55, 66].

## 5.2 Test of decay constants

We have just outlined a proposed method for computing the localisation length at very low energies. Let us assume for the moment that the decay model we have developed applies (i.e. that the eigenstates take the form of one or a handful of strongly occupied domains with straight-forward decay through the valley lines into their neighbours). Under these conditions, the domain area calculation can hardly fail to give $D$ correctly, the mean distance separating tunnelling-inducing valley lines. On the other hand, the decay constant from one domain to another, $\bar{\rho}_E$, is a different matter entirely. As will be discussed in section 7, the level of approximation involved is very high, and there is no *a priori* assurance that our method yields numbers which faithfully capture the decay of the eigenstates. Therefore, a direct test is in order. This can be done as follows: for the same noise realisation, we perform the full LLT calculation, as well as find the low energy eigenstates by exact diagonalisation. Now, we know that within each domain, the wavefunction remains roughly constant (same order of magnitude). Therefore, we integrate $|\psi|$ over the domains, and divide by the domain areas to get the average of the wavefunction amplitude on each domain.

Then, by visual inspection of the eigenstates, we find examples of eigenstates and domain pairs where it is clear that the wavefunction tunnels from one domain to the other, as opposed to an independent occupation of the two domains (or any of the more complex behaviour described in section 6 which is encountered at higher energies). We also avoid higher local modes than the fundamental (excited local states involve nodes of the wavefunction within a domain). Having identified suitable candidates, we take the ratio of the mean amplitudes on the two domains and compute the logarithm. The resulting number is equivalent to $\bar{\rho}_E$ from LLT, the exponential cost of going specifically between these two domains (in this noise realisation), at an energy equal to the eigenvalue corresponding to the eigenstate examined.

We have performed this test, and the results are shown in the top panel of Fig. 7. A clear correlation is seen, whether the predictions of LLT are compared to the eigenstates of $H$ with potential $V$ or $W_E$. The performance of the LLT method is equally good for arbitrary strengths

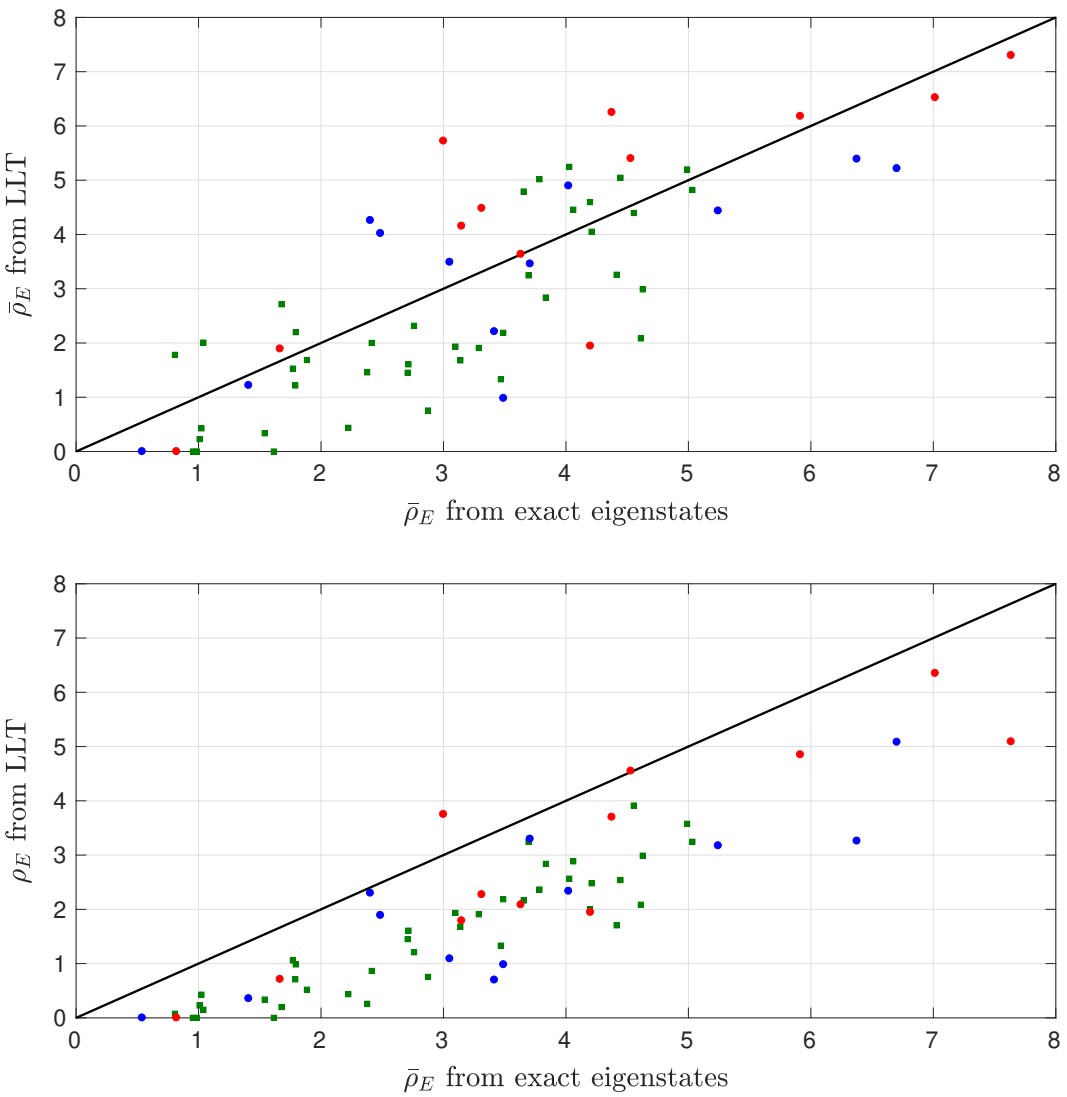

Figure 7: Exponential decay cost linking two neighbouring domains, plotting the values measured from exact eigenstates and LLT against each other. Top: the LLT calculation shown uses the average over all linking paths between domain pairs, and there is a very clear correlation to the eigenstate decay coefficient: the data points fall nicely around the identity map, shown as a black solid line. Bottom: data for the exact same test-cases is now plotted so that the LLT calculation shows the minimal path only over all linking paths between domain pairs, and it is obvious that this method systematically underestimates the true decay rate. All data points presented were obtained for a system with $L = W = 25\ell$, $V_0 = 21.33E_0$, $\sigma = 0.48\ell$. Blue and red circles have $f = 0.02$, with blue coming from diagonalising $H$ with $W_E$ and red with $V$, while green squares used the real potential $V$ and $f = 0.1$.

of localisation (compare sparse and dense scatterer results), simply because the only numbers included in the test are those for which the eigenstates and domains chosen are sensible (sufficiently low energy, correct local modes, decay as opposed to independent occupation, etc.). Of course there is scatter about the identity function, but since much averaging is performed during the calculation of $\xi_E$, this scatter will disappear in the mean. This gives us confidence in the validity of our novel computational method for very low energies.

In contrast, as mentioned earlier, the Agmon distance itself, $\rho_E$, systematically falls short of the true decay coefficient (being a formal lower bound), as depicted in the bottom panel of Fig. 7.

We emphasize that there is no other available method to compare our calculation of the localisation length to. The only reliable approach is to run time-dependent simulations, integrating the Schrödinger equation. The simplest test would be to initiate a translating Gaussian wavepacket with a fairly narrow energy distribution outside the disorder, allow it to propagate, and observe the resulting exponential decay set in with time. The (unnormalised) energy distribution for our translating 1D Gaussian initial condition is simply

$$g(E)\, dE = \exp\left[-2\bar{\sigma}^2\left(\sqrt{\frac{2mE}{\hbar^2}} - k_0\right)^2\right] \frac{\sqrt{m}}{\hbar\sqrt{2E}}\, dE. \tag{11}$$

One would have to average 20 to 30 realisations to get accurate results, measure the decay length scale seen in the density, and compare to that obtained from the energy-resolved $\xi_E$ obtained from LLT by reconstructing the expected density profile for the given energy distribution according to, e.g., equation (63) of Ref. [2]. However, taking into account the energy distribution of the wavepacket would in this case only provide a fine-tuning of $\xi_E$ taken at the mean energy of the narrow Gaussian wavepacket, which would provide a very good estimate already.

We have attempted precisely such testing of our LLT $\xi_E$ (in parameter regimes and at low enough energies where the curve $\xi_E(E)$ is smooth and monotonically increasing), to find that the LLT prediction greatly *underestimates* the real localisation length, by up to as much as an order of magnitude. For example, using a system geometry given by $L = 50\ell, W = 25\ell, R = 30\ell$, noise with $f = 0.1, \sigma = 0.48\ell, V_0 = 21.33E_0$, initial condition specified by $\bar{\sigma} = 5\ell, k_0 = 1/\ell$, and evolving the state for a total time of $100t_0$, we find that quasi-steady state in the density profiles is achieved at $\sim 40t_0$, after which the exponential profile changes slowly and can be meaningfully fitted. We extract a time-dependent localisation length from the density profiles, $\xi(t)$, which increases from $10\ell$ to $13\ell$ over the fitted time interval ($60t_0$) of the simulation, and would only increase further with time before eventually equilibrating to a constant. Meanwhile, the LLT calculation, combined with equation (63) of Ref. [2] and the energy distribution (11), together with an exponential fit to the overall predicted density profile, yields a value of $\langle \xi \rangle \approx 2.746\ell$, which is considerably smaller.

The reason for this discrepancy is that the simple decay model that we have been assuming is only valid at very low energies, after which more complex mechanisms of how the eigenstates can spread out spatially come into effect. These are beyond quantum tunnelling and the semiclassical theory thereof, and are described in detail in section 6. In the example above, the energy distribution lay fully outside of the applicability regime of the LLT calculation. Comparison to time-dependent simulations in a regime where the simple decay model applies are further discussed in section 6.

## 5.3 Effect of parameters

Let us consider – and when possible, examine – the effect of the different parameters in the model on the localisation length obtained via the prescription given in this section. Firstly, the

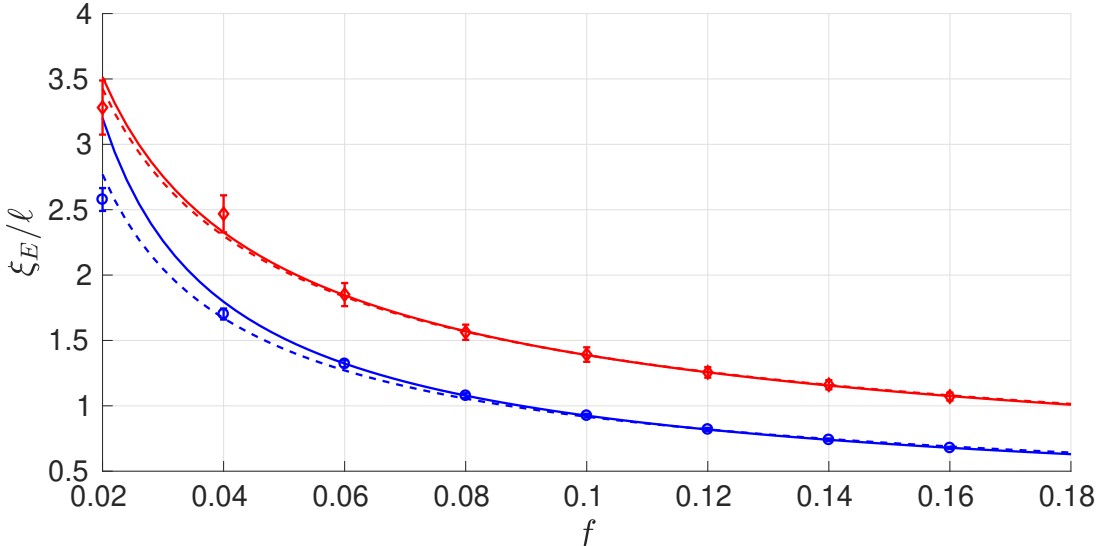

Figure 8: The eigenstate localisation length $\xi_E$ at zero energy as a function of fill factor for two scatterer heights, blue circles: $V_0 = 21.33E_0$, red diamonds: $V_0 = 5E_0$. For all simulations $W = 25\ell$, while $L = 125\ell$, $\sigma = 0.48\ell$ for the data shown in blue and $L = 25\ell$, $\sigma = \ell/2$ for the data plotted in red. The lines are fits according to equation (13), with the colour matching the data set being fitted. Dashed lines fit all the data points, while solid lines exclude the first two (as it increases the fit quality and it is possible that the lower fill factor points are not very accurate). Error bars show the standard error. The effect on $\xi_E$ of using different system sizes in the two cases is about $0.03\ell$ (using $L = 25\ell$ for the $V_0 = 21.33E_0$ data set increases $\xi_E$ by about $0.03\ell$), which is an order of magnitude smaller than the effect seen in the figure. Thus, increasing the fill factor or scatterer height decreases $\xi_E$, as expected.

calculation can be performed as a function of energy, and as expected, the computed number increases with energy monotonically until one reaches the regime where the finite extent of the system limits the calculation and artificially reduces $\xi_E$, as well as the mobility edge predicted by LLT but found unphysical in [55, 65], beyond which it is no longer possible to perform the calculation. However, the computation ceases to be valid much earlier than that, because the pure decay model we assumed breaks down, as illustrated in section 6. In fact, it is usually only very low energy eigenstates that are captured correctly by our description, and the only method known to us of establishing when the complex decay behaviour (section 6) begins is by visual inspection of the exact eigenstates. This "complex decay" is beyond quantum tunnelling and semiclassical theory, and is attributed directly to Anderson localisation. We will therefore only show data for $E = 0$, where the results have been confirmed as meaningful across the range of parameters shown.

Figure 8 demonstrates that the localisation length is reduced by strengthening the disorder by either increasing the scatterer height or the fill factor. Increasing the width of the scatterers also decreases the localisation length, but we do not simulate this directly in this paper. System size only influences the results weakly due to finite size effects studied thoroughly in [55, 65].

Since we have the opportunity, we compare our results to the analytical formula for the localisation length in 2D

$$\xi \sim \ell_B \exp\left(\frac{\pi}{2} k\ell_B\right), \tag{12}$$

where $\ell_B$ is the Boltzmann mean free path and $k$ the wavenumber associated with the energy at which the localisation length is evaluated. The Boltzmann mean free path (the distance

over which the wave loses memory of its initial direction) is related to the scattering mean free path $\ell_s$ (the mean distance between scatterers) through the scattering cross section of a single scatterer, which includes information about the scatterer height and shape, as well as the energy of the wave. We recall that while this formula is quite freely used in the literature (e.g. [15]), it is not expected to be correct, as it is derived (for a classical wave) by first assuming weak localisation and then forcing the diffusion coefficient to zero [4,5] (in addition, we do not have white noise or an infinite system).

One may relate the mean free path to the fill factor rather trivially by simple geometrical arguments, yielding $\ell_s \propto 1/\sqrt{f}$, and then fit the numerically-obtained $\xi_E$ as a function of fill factor to

$$\xi \sim \frac{a}{\sqrt{f}} \exp\left(\frac{b}{\sqrt{f}}\right). \tag{13}$$

This has been done in Fig. 8, and the fits are of reasonable quality. However, this does not prove the validity of equation (12), as one would have to check the energy dependence of the fit coefficients for consistency with the formula, an impossible task in our case since the LLT calculation is limited to such low energies.

## 6  Breakdown at higher energies

As we have briefly mentioned in the previous section, the localisation length extracted from time-dependent simulations disagrees with our LLT prediction, even when we limit ourselves to sufficiently low energies where $\xi_E$ is smooth and monotonically increasing. In fact, the localisation length from time-dependent simulations is considerably larger, by up to as much as an order of magnitude. In this section we explain how and why this occurs, based on an analysis of the structure of the eigenstates, using Fig. 9 for illustration. In particular, we find that the pure decay model (applicable for example to the first eigenstate in Fig. 9) we have assumed thus far ceases to be relevant beyond very low energies, and describe the mechanisms by which the wavefunction spreads out across the system that come into play at higher energies. These effects are beyond quantum tunnelling and its semiclassical approximation, violating the Agmon inequality (8), and are best ascribed to Anderson localisation directly.

We have already explained that as the energy increases, the valley lines of LLT cease to be effective and domain walls break open, as segments of the potential barriers between them are "submerged". When the breaks in the domain walls are small, one still sees some exponential decay through such walls (e.g. third eigenstate in Fig. 9, decay from second to fourth domain), even though semiclassically (according to the formal Agmon distance), it is now possible to go across the barrier at no cost at all. In this low-energy regime, our use of $\bar\rho_E$ to capture the tunnelling *and* base the domain area merging on its vanishing is sensible. However, as the gaps in the domain walls grow, it becomes common to have single-amplitude bumps extending between domains through these gaps (e.g. sixth eigenstate in Fig. 9, between the fourth and eighth domains), and one can no longer talk of decay. In this regime, it would be better to use $\rho_E$ proper (which indicates that no tunnelling occurs) together with the criterion $\rho_E = 0$ to merge domains. This is one mechanism that causes the true localisation length to be greater than the one we compute. Since there are many others (see below) that make a sensible calculation of $\xi_E$ at higher energies impossible anyway, we choose to persist with $\bar\rho_E$, which is the correct number to use at low energies.

A prominent, strongly dominant mechanism going beyond the pure tunnelling picture is what we shall term the "seeded excitation" scenario. Here, ordinary tunnelling from a strongly occupied domain into its neighbour excites a local mode inside that domain (usually manifesting as a separate bump), of an amplitude set by the decayed wavefunction in the "receiving"

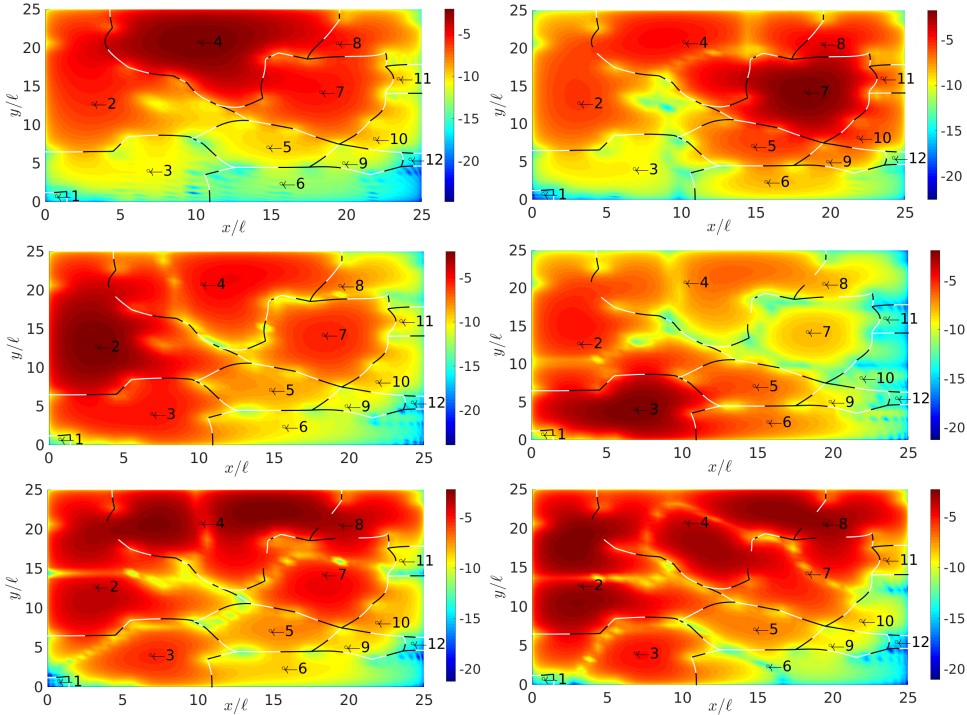

Figure 9: Six of the lowest energy eigenstates (going across the panels and then down) for a single noise realisation in a system with $L = W = 25\ell$, $V_0 = 21.33E_0$, $\sigma = 0.48\ell$, $f = 0.06$. The colour map depicts the logarithm of the amplitude of the wavefunction. Black and white lines (the different colours are used simply to make the structure of the valley network clearer) show the effective domain walls (where $E < 1/u$) at the given eigenstate energies, having removed any open valley lines first. Maxima of $u$ are marked with open black circles and the corresponding domains are numbered for ease of reference. The eigenstates demonstrate the various possible mechanisms by which the wavefunction can spread across the system, other than by pure decay (see discussion in the text). Once these mechanisms come into effect, our calculation of the localisation length loses meaning; this constrains the regime of its applicability to very low energies.

domain. Many examples of this can be seen in Fig. 9, with the lowest-energy case occurring in the second eigenstate, going from the seventh and eighth domains into the fourth, as well as the fourth to second (although here the local excitation and the original decayed amplitude are merged and it is the amplitude maximum in the second domain that is the tell-tale sign of seeding). The effect of seeded excitation is to strongly increase the weight of the eigenstate on the "receiving" domain (that is, increase the average value of the wavefunction on this domain), and as a result, decrease the decay coefficient between the domain pair in question.

Another mechanism that comes into play at higher energies is "resonant excitation". Occasionally, we find domains excited without any significant decay into them from other, strongly occupied domains (e.g. seventh domain in the third and fourth eigenstates of Fig. 9). In such cases, the excitation is caused by "resonance" with a mode in a near-by occupied domain (in the examples provided, it is probably the fourth domain which is responsible). Note that such resonances can happen even between domains that are of considerably different areas, as long as higher modes are involved, so that the *mode* energy is close. This scenario allows the eigenstates to cover a larger area without undergoing a decay. As energy increases and higher mode

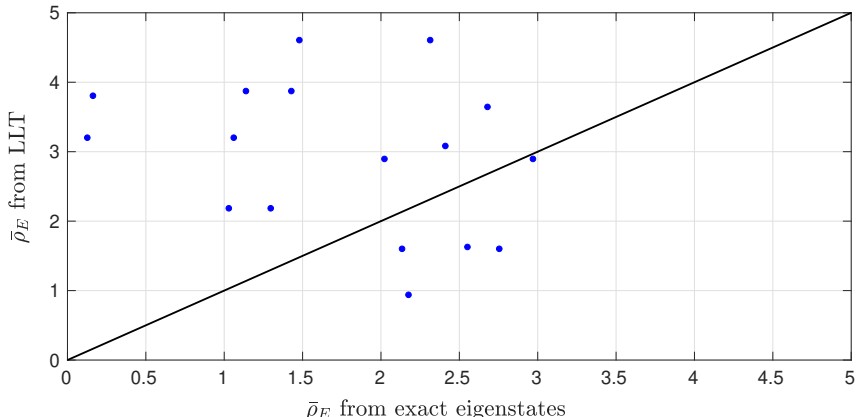

Figure 10: Exponential decay cost linking two neighbouring domains, plotting the values measured from exact eigenstates and LLT against each other. Data points were obtained for a system with $L = W = 25\ell$, $V_0 = 21.33E_0$, $\sigma = 0.48\ell$, $f = 0.06$. Only cases that do *not* fit the clean decay model tested in Fig. 7 were included in this test, and we see that overall, the LLT method yields a larger decay coefficient than the true value due to all the additional complex mechanisms, effective at higher energies, discussed in section 6.

excitations become more prevalent, more and more resonances are possible as the range of available energies to match grows.

An interesting observation regarding resonant excitations is that the distance between the two domains in question is never very large (perhaps a gap of two or three domains at most), so that overall, the occupied domains are still clustered and the states are localised. A possible explanation may be that intermediate detuned domains reduce the coupling between the resonant domains, which only allows fairly local resonant excitations.

Clearly, higher-order modes (e.g. Fig. 9, the fourth domain in the fifth eigenstate has a prominent node) are not accounted for in our description of section 5, but this is not a serious problem, as usually, all the bumps within a single domain have similar amplitudes and the nodes between them do not reduce the mean value of the amplitude on the domain by much.

The final complicating factor is that even simple decay can occur from several nearest neighbours (e.g. Fig. 9, in the sixth eigenstate, the seventh domain gets a contribution from both the fourth and eighth domains), which implies that the mean value of the wavefunction on that domain will be greater than it would have been if only one such decay contributed to its population.

All (but the last) of the factors outlined so far are beyond quantum tunnelling, violate the Agmon inequality (8), and should be thought of directly as quantum interference effects. They serve to increase the localisation length beyond the value calculated according to our LLT method, which is therefore only valid at very low energies, for maximally-localised states. This happens due to both larger effective distances separating decay events, which is fairly straight-forward to both understand and visualise, and weaker decay when such events do occur. The latter has been quantitatively confirmed by comparison to exact eigenstates (Fig. 10), this time choosing domain pairs that do *not* fit the pure decay model, but involve one or several of the more complex mechanisms discussed in this section. It is clear that LLT overestimates the decay coefficient, and the data shown can certainly accommodate the observed difference between LLT and time-dependent simulations in a regime where these mechanisms are prevalent. Considering the contribution from the larger effective area (compared to that assumed by the pure decay model of LLT) which also serves to increase the localisation length, these

results are consistent with and explain why density profiles from time-dependent Schrödinger simulations indicate a larger localisation length than that predicted by LLT.

We note that it is possible to set the disorder strength so high that the pure decay model is applicable up to sufficiently high energies (confirmed by examining eigenstates) that a slowly translating Gaussian can fit in to the range of energies where our LLT calculation should show agreement with time-dependent simulations. We have done this test, but found that once again LLT underestimates the localisation length extracted from time-dependent simulations. For example, in a system with $L = 25\ell, W = 25\ell, R = 30\ell$, noise parameters $f = 0.2, \sigma = 0.48\ell, V_0 = 21.33E_0$, a translating Gaussian with $\bar{\sigma} = 5\ell, k_0 = 0.5/\ell$, and total evolution time of $100t_0$, quasi-steady state is reached at $\sim 50t_0$, after which the fitted localisation length $\xi(t)$ stays roughly constant at the average value of $8.3\ell$. On the other hand, the localisation length predicted by LLT is $\langle \xi \rangle \approx 0.62\ell$, indicating much stronger localisation. In this case, the LLT calculation is valid over the entire range spanned by the energy distribution of the wavepacket used.

This can be explained by "edge effects": for the direct time-integration of the Schrödinger equation, one needs empty, noise free "reservoirs" on either side of the disordered system to initiate the wavefunction in and to collect any transmitted atoms. The coupling of the noisy region to the empty reservoirs modifies the valley network in the vicinity of those edges of the system, as the wavefunction is not forced to zero by boundary conditions, but is allowed to take on arbitrarily high values. In particular, localisation is weakened through the fact that some of the valley lines disappear upon addition of the reservoirs, and the barriers in the effective potential along the remaining valley lines are lower as $u$ near the "coupling" edges is higher. When the disorder is so strong that the ("internal system") localisation length, as computed from LLT, is smaller than the typical domain size, these edge effects become important. Strong decay happens over one or two domains, and because in the time-dependent simulations, it is precisely the affected regions that the wavefunction passes through as the exponential decay sets in, we observe a larger localisation length than is found away from the edges. This explains why we still see a discrepancy in the regime where the LLT method should in principle work well.

Now, an interesting observation is that it is not only the lowest energy eigenstates in $H$ with $W_E$ conforming to the pure decay model which are similar to those in $H$ with $V$, but also a few modes beyond this regime, when quantum tunnelling in $W_E$ is already insufficient to capture the structure of the eigenstates. This can be seen in Fig. 5, for the second to fifth eigenstates. The reason that a few low energy modes are similar even in the regime where the tunnelling model is no longer applicable is simply the similarity of the two potentials, $V$ and $W_E$, the latter being a smoothed version of the former. While the correlation between the eigenstates in the two landscapes is lost at higher energies, all the mechanisms discussed in this section can also be seen in the eigenstates of $H$ with $W_E$ beyond that point.

The understanding that the pure tunnelling picture in LLT is only applicable at very low energies is novel, and establishes when and how one can use LLT in a useful manner. Two very recent papers [67, 68] have developed generalisations of LLT to allow treatment of systems with internal degrees of freedom, with [68] explicitly extending their technique to arbitrary energies, while the method in [67] is amenable to such an extension [68]. These generalisations of course come at the price of added complexity, but have additional advantages as well: for example, the method of [68] removes the constraint that the physical potential cannot be negative on any part of the system domain, and yields some helpful features arising from the different normalisation of the eigenstates chosen therein. On the other hand, it should be noted that the paper [68] has only presented examples of their calculation performed at zero energy.

# 7 Multidimensional tunnelling

The Agmon distance of LLT (7), including minimisation over all paths connecting the two points in space, gives a prescription to predict the minimal decay of eigenstates through the barriers of $W_E$ as they tunnel out of each domain – a local potential well – and spread across the system. In section 5 we have heuristically outlined and tested a method to quantitatively estimate $\rho_E$ between neighbouring domain minima of $W_E$, avoiding the path minimisation stage, but using the usual expression for the integrand along the path.

Multidimensional tunnelling is in fact an old and thoroughly-investigated problem. Of course, brute force quantum mechanical calculations are possible, but physicists have been striving to obtain *insight* into the process by generalising the WKB approximation to dimensions higher than one to describe it. In 1D, WKB is a straight-forward and methodical approach (see, e.g., [69]) – a controlled approximation that is fully understood. The generalisation to several dimensions is a different matter entirely: there is a large body of literature developing and discussing different methods, their limitations, suggesting improvements, and utilising these techniques to solve practical problems. In this section, we will provide an overview of this topic, to place our method of section 5 in perspective.

Let us see where the Agmon distance equation (7) comes from. The starting point of the derivation is usually the Feynman propagator, none other than the Green's function of the system. One has to go through a series of approximations, listed below, in order to arrive at this semiclassical formalism:

1. The propagator is expanded in powers of $\hbar$, and only the zeroth order term is retained[3] [70, 71].

2. Next, one usually assumes that Hamilton's principle function is pure imaginary [70, 72, 73].

3. In principle, if we want to use the Feynman propagator to describe tunnelling from one region of space where the wavefunction is initially contained to another, we must consider all source points, all target points, and all possible paths to arrive from each source to each target point. In the simplest approximation, one uses the fact that the contribution of the classical path is the largest, and as we move away from it in configuration space, the contribution of the other paths is exponentially suppressed. Therefore, one usually only examines the classical path, or at most a "tube" of paths around the classical one. Moreover, it is common to only consider one source point (at which the wavefunction is maximal) and one target point (say the minimum in the potential on the other side of the barrier). The classical trajectory method was developed and used in many papers, e.g. [71, 74–76], and relies on minimising the action via the Euler-Lagrange equations.

Assumption 1 is already a strong limitation, and to the best of our knowledge, first order solutions were only ever obtained in the classically allowed region [70]. However, taking $\hbar \to 0$ is the essence of the semiclassical nature of the method, and not much can be practically done to overcome this approximation.

Assumption 2 is certainly not generally justified [70, 72, 73]. These three references have superbly dealt with the case of a general complex action, and demonstrated that a geometrical ray construction, following two surfaces (equi-phase and equi-amplitude) along two orthogonal paths, is necessary to solve the problem in earnest. They have proven that the imaginary action approximation breaks down if one considers a general incoming wavefunction, incident on a barrier such that its $k$-vector is arbitrarily predetermined. It has also been argued that

---

[3]An equivalent approach is to write the wavefunction in polar form and expand the phase similarly.

this approximation can even fail for tunnelling out of a potential well [73]. The geometrical construction proposed in these papers is extremely involved, and is completely impractical for our purposes.

While in principle, accuracy could be improved by including more than one source and target point, as well as considering multiple paths as in [71], all three simplifications of the third assumption are essential for our case: we cannot afford (computationally) to calculate many paths or to describe each domain by anything more than the point at which $W_E$ attains its minimum. This is because the calculation needs to be done so *many* times that it is simply impractical.

The usual final form of the semiclassical approximation in the forbidden region involves solving the classical equations of motion with negative the potential and the energy, or equivalently, in imaginary time. The differential equations are based on Newton's laws, imposing energy conservation as a constraint, and seek out the path of minimal action. In 2D, they take the form

$$
\begin{aligned}
\frac{d^2 x}{ds^2} &= \frac{\frac{\partial V}{\partial x} - \frac{dx}{ds}\left(\frac{\partial V}{\partial x}\frac{dx}{ds} + \frac{\partial V}{\partial y}\frac{dy}{ds}\right)}{2(V - E)}, \\
\frac{d^2 y}{ds^2} &= \frac{\frac{\partial V}{\partial y} - \frac{dy}{ds}\left(\frac{\partial V}{\partial x}\frac{dx}{ds} + \frac{\partial V}{\partial y}\frac{dy}{ds}\right)}{2(V - E)}.
\end{aligned}
\tag{14}
$$

Here, $s$ is the arc length along the path defined by the coordinates $(x, y)$, and $V$ is the potential the particle with energy $E$ moves in. In this parametric form, the equations for the allowed and forbidden regions are identical.

In the context of tunnelling out of a potential well, the trajectory is usually required to pass through the turning surface (where the kinetic energy vanishes) normally, so that it can connect smoothly to a classical trajectory in the allowed region. On the turning surface, the velocity is aligned along the gradient of the potential [71, 76]. An alternative constraint was used in [74]: the authors required their escape paths to pass through the saddles of the potential and be aligned along the correct axis of the saddle at those points (which is closer in spirit to our approach, but is less rigorous). Essentially, if the direction of the incoming wave is predetermined and it impinges on the turning surface at any angle other than normally, the action must be taken as complex and the classical equations are insufficient. This is the chief difference between tunnelling out of a local well and the transmission of an incoming wave through a barrier.

We highlight that in the final form of the semiclassical approximation, the minimal path is energy-dependent: one must solve the set of ordinary differential equations defining the minimal path for each energy separately. If we wish to find the classical path that connects two specific points, knowledge of the energy gives us the magnitude of the velocity vector, but its direction is unknown. Trial and error is called for to discover the latter: one needs to try different initial directions of motion until a path that arrives at the desired end point is found. Furthermore, if the two points of interest are separated by one or more turning surfaces (which cannot be crossed classically), one must begin at each of the two points, and try different directions until a trajectory that hits the turning surface normally and is reflected back on to itself is found. In order to connect points lying on turning surfaces, in principle, the initial direction of the velocity can be found from the gradient of the potential, but in practice, fine-tuning is still necessary. Thus, in general, finding the true classical path is a piece-wise process and takes multiple rounds of guessing the direction of the velocity. This makes the traditional (and formally correct) solution of the semiclassical problem impractical for our purposes.

Our method of section 5 overcomes these difficulties: no differential equations need to be solved at all (one only needs to know the localisation landscape $u$), one path is computed

for all energies, and there is no need to guess the initial condition. As we have seen in Table 1, it performs well, which justifies its use despite the many approximations in deriving the semiclassical formulation, as well as our heuristic way of computing the escape paths. In either case, no other level of approximation is practical for our purposes, as we need to compute the "mean" Agmon distance between every two neighbouring domains at all energies for many noise realisations (twenty are used in practice), at each set of parameters investigated.

The additional discovery that by averaging over all candidate paths of LLT, the mean Agmon distance $\bar{\rho}_E$ in fact yields the average decay rate (rather than the minimal) is a further point of merit to our approach. The other candidate paths cannot be obtained from the rigorous semiclassical formalism, which so far has only been able to provide a lower bound on the decay rate.

A few final notes are in order, without which any review of multidimensional tunnelling would be incomplete. References [77, 78] have developed the path decomposition expansion method, which allows one to divide space into separate regions, minimise the action in each one using whatever method happens to be optimal in that vicinity (chosen based on physical considerations), and then collate the solutions using global consistency equations. Reference [71] deserves special attention, as an exceptional effort was made to consider many classical paths from many source points, deriving the tunnelling current and transmission coefficient through the potential barrier.

For a more comprehensive review of the topic, the reader is referred to [79], as well as the original literature cited above.

# 8 Conclusions and future work

In this paper we used LLT to calculate the eigenstate localisation length at very low energies, quantifying the decay length scale of the eigenstates. This required us to develop a practical approximation to multidimensional tunnelling and a formidable extension of LLT techniques and machinery. It also involved considerable conceptual progress, linking together domain size and the decay exponent (the "cost") of tunnelling through the peak ranges of $W_E$ separating domains through the saddle points. We accounted for the effect of increasing energy by merging domains as the domain walls separating them broke down. Crucially, we explicitly tested the decay coefficients computed from LLT against exact eigenstates, validating our computational method and the many approximations involved. We improved on the direct use of the Agmon distance, which gives a lower bound for the true decay coefficient, and found a way of computing the latter in a way that avoids a one-sided bias. We also reviewed multidimensional tunnelling to set our method in context.

We gave a thorough discussion of how the eigenstates spread out over larger areas at higher energies, beyond the regime where quantum tunnelling in $W_E$ and its semiclassical description is applicable, and explained why the mechanisms involved are not captured by our method for computing the localisation length, thus necessarily limiting it to very low energies. In addition, we highlighted the difficulty in extracting the localisation length out of exact diagonalisation calculations. We further demonstrated that the effective potential $W_E$ can replace the real potential $V$ in the Hamiltonian in terms of reproducing the low-energy eigenspectrum.

Some ideas for future work that naturally came up during this investigation are:

1. It may be possible to perform the LLT calculation of $\xi_E$ for a system where the Green's functions approach would be applicable, even if it would only provide approximate results, and compare the two.

2. It would be excellent to generalise LLT to 3D, where the logic and conceptual picture are

largely unchanged, but the practical framework and the technology are not yet in place (everything beyond obtaining $u$ and performing simple mathematical operations on it). This would open the door to a large number of possible studies in 3D.

3. One should also investigate the functional dependence of $\xi_E$ (at zero energy) on the fill factor and $V_0$. At the moment, this can only be done by running large numbers of simulations at different parameters and examining the dependence explicitly, hoping to discover the functional form by inspection.

4. What effect does the shape of the scatterers have? We have limited ourselves to 2D Gaussian peaks (of more or less constant width) for this paper. What would happen if we changed the width, or even made the scatterers, say, square?

# Acknowledgements

S.S.S. warmly thanks the following researchers for extremely helpful discussions on the topics indicated in parentheses after each name: Daniel V. Shamailov (the entire project), Antonio Mateo-Munõz (spectral methods in exact diagonalisation), Xiaoquan Yu (importance of the density of states for Anderson localisation), Marcel Filoche and Svitlana Mayboroda (the Agmon distance). Jan Major is further gratefully acknowledged for reading the manuscript and providing useful comments.

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
