# Peer review of "Computing the eigenstate localisation length at very low energies from Localisation Landscape Theory"

_SciPost Physics Core, doi:SciPost Phys. Core 4, 017 (2021)_

## Round 1 · Referee Report · Anonymous (Referee 1) · 2020-9-11

Strengths

1 - calculation of localization length based on localization landscape theory
2 - proposed simple algorithm to compute approximate Agmon distance
3 - good literature review

Weaknesses

1 - lack of comparison of the estimate of Agmon distance with its real value
2 - lack of comparison of obtained localization length with established methods
3 - lack of comparison of localization length obtained from $W_E$ and $V$

Report

In this paper authors discuss how the effective potential $W_E$ obtained from the localization landscape $u=H^{-1}\mathbf{1}$ can be used in place of the real disordered potential $V$ in finding the properties of solutions to the Schrodinger equation. The main new result of the paper is a method of computing localization length based on the effective potential $W_E$. While the idea that $W_E$ can govern the decay of eigenstates through associated with it Agmon distance $\rho_E$ was already proposed in one of the original LLT papers [47], this is, to my knowledge, the first attempt to use this idea to compute localization length in practice. Authors present a convenient way of finding the approximate Agmon distance, which otherwise would require finding minimum over infinite number of possible paths. They also propose a new formula to calculate an estimate of localization length based on the Agmon distance computed in such a way.

While these are interesting results, the paper mostly lacks in comparing them with other established methods.

Firstly, authors only compare the approximate Agmon distance between two basins of $W_E$ with the decay of wavefunction between these basins (Fig. 8). While these two result show a good correlation, which is in favor of the proposed method, whether the estimate of Agmon distance is a good one still remains in doubt. It should be shown that computed estimate agrees well with the exact value. This could be done, for instance, for several values of energy $E$ and a couple of pairs of minima of $W_E$. Authors mention in section 6 that this is impractical when the calculation has to be performed so many times, however I think it should still be feasible for the sake of such demonstration.

Secondly, the computed values of localization length are not compared with results obtained with any established method, such as based on transmission coefficient as in Ref. [26]. This comparison is necessary to determine, whether the proposed method is actually useful.

Thirdly, as authors notice in section 4, replacing $W_E$ with $V$ works better as the strength of disorder increases. However, as the strength of disorder tends to infinity we have $W_E \approx V$. This raises a concern that the authors might be working in the regime where $W_E \approx V$, when computing quantities based on $W_E$ gives no advantage over simply using $V$. This concern would be avoided if authors also provided results of localization length based on $V$ as well as $W_E$, which would hopefully show that those based on $V$ are meaningless. Paper would also benefit if authors included a figure presenting a typical realization of disorder $V$ and the corresponding effective potential $W_E$ to demonstrate that the two are in fact qualitatively different.

In my opinion, this work has the potential to constitute an important development in the field of localization landscape theory, by making use of its ideas to calculate meaningful physical quantities. At this point, however, it is not clear whether this calculation is actually valid, as I expressed in the three points above. If these concerns are addressed, and the results hold up to such scrutiny, I would recommend this paper being published in SciPost Physics. Otherwise, it would significantly diminish interest of this work.

Requested changes

1 - Exact value of Agmon distance should be calculated for a couple of examples and shown whether it agrees with the proposed estimate 2 - Localization length should be calculated using established methods, such as bases on transmission coefficient (e.g. Ref [26]) and compared with the results obtained using the method proposed by the authors 3 - Localization length should be computed using Agmon distance based on $V$ (or it should be shown that such calculation makes no sense), to determine whether there is an advantage of using $W_E$ over $V$ 4 - A typical realization of disorder $V$ and the corresponding effective potential $W_E$ should be presented to determine whether they are qualitatively different

---

## Round 1 · Referee Report · Anonymous (Referee 2) · 2020-9-13

Strengths

1-A potentially interesting and powerful way of calculating the localization length from the localization landscape is provided

Weaknesses

1-Missing more direct comparison with earlier studies of localization lengths with more traditional methods

2-Some conclusions and discussions could be more quantitative, while at the moment a qualitative discussion bases on a handful of data curves is given

3-The motivation for much of section 4 is unclear

4-The paper is not self-contained

Report

Before discussing the contents of this paper, I have to say that the origin of this paper is somewhat unusual. The authors have posted a long (111 page) paper on arXiv (2003.00149), which they call a report, and state there that they plan to submit the contents of that report as five shorter articles to SciPost. It is unclear if they plan to, or are in the process of publishing the longer report, in which case it would seem that the shorter articles could not be published as original and new work.

This way of writing up their work in two steps is clearly felt in this article. In several places, the manuscript refers to the longer work (Ref. [52]) for details, often details that are essential for reproducibility and understanding of what is done in this paper, and are otherwise not contained in the paper. Sentences like "Details about our computational techniques and the numerical methods employed can be found in the the appendices of [52]," "a more detailed description of the system can be found in [52]," and "see appendix C of [52] for details" are found throughout the text. This makes the paper not self-contained, as it is not possible to even know what has been done without having to refer back to the longer report. There are more signs of this being taken from a longer report: before Eq. (6) the authors write "we repeat here the definition of the energy-dependent quantity known as the Agmon distance," despite this quantity never having been given earlier in the manuscript. I assume it was given somewhere earlier in Ref. 52. For a paper to be publishable in SciPost it needs to contain new results and to be self-contained so that one can read the paper and understand what was done without having to read another paper. It is not just details that we are talking about here, it is, for example, essential information on how the data being presented was obtained that one can not extract directly from this paper.

This brings me to the actual contents of the paper. The authors start by giving arguments, based on their exact digaonalization data, for why it can be hard to extract the localization length from exact eigenstates. I don't think there is anything that is really new in this part as it is known that extraction of localization length can be a nontrivial task, though in some cases there are good methods for doing this. This, however, motivates the problem that they want to address, namely using the localization landscape to extract a localization length.

In section 4 they discuss the effective potential $W_E = 1/u$ with $u$ the localization landscape obtained by solving $Hu = 1$. I think this section is supposed to be here to motivate that $W_E$ is a good potential to describe localization, something that has been done in several papers before. However, I fail to understand the logic and motivation for the authors approach. In order to try to show this to be the case, the authors decide to solve the Schrödinger equation with $W_E$ as a potential and compare the eigenstates and eigenenergies with those obtained with the potential $V$. They don't really give any motivation for why this should be a meaningful comparison. In my understanding, $W_E$ works nicely to describe localised states because somehow in the construction of it, quantum interference and resonance effects are taken into account. $W_E$ can then be used essentially as a classical or semi-classical potential. It doesn't make sense, to me at least, to reinsert this into the Schrödinger equation, since that is a completely different problem from the original one, and it will add quantum interference effects and resonances into the problem that where not originally there. And if one is solving the Schrödinger equation, then surely one would just solve the one with the actual potential $V$. I may have missed something here, or things have not been explained clearly, but if not I fail to understand what one can actually learn from this calculation.

Section 5 is the most interesting section of the paper. This is based on the Agmon distance given in Eq. (6) that was already introduces in Refs. 47 and 48, where it was shown to "control the decay of the wave function in regions where $E < W$." That is, the exponential decay of the wave functions comes only from regions where the effective potential is higher than the energy. Here the authors use this to extract a localization length. In order to achieve this the make the simplifying approximation that it is sufficient to consider only nearest neighbour domains in the effective potential and to estimate the localization length from the decay of the wave function, as obtained from the Agmon distance between these two domains, via Eq. (8). This seems like a rather strong approximation that one would really only expect to hold for states that are very localised and at most have significant support on two domains. The authors say as much, and relate that to the energy of the eigenstates, which they claim should be small in order for this to work. It is an interesting question, not solved in this paper, how one would use the Agmon distance for more general eigenstates to extract localization lengths. The authors discuss in the last section the connection of the problem to multidimensional tunnelling, though without any suggestions on how to go beyond what they have done.

I am not fully convinced about how successful this method has been for the authors in the example they have studied. Figure 8 compares the localization length from exact eigenstates and from the localisation landscape. A solid line representing when they agree is drawn through the data and the authors claim that there is good correspondence between the two. I don't know, there is some correlation but the variance is also rather high. It would have been good to have some systematic way of estimating more quantitatively how well their method works. Either by applying their method to systems where some more results are known, or by having a way to improve on the approximation they make in extracting the Agmon distance, for example by going to further domains as well.

I also worry a bit about the system sizes that are being looked at and the localization lengths being extracted. If I look at Fig. 9 and Fig. 10 I notice that the localisation length is a significant fraction of the system size, and for much of the data actually even larger than system size. It's hard to imagine one can get reliable localization lengths in this case. If I look at Fig. 7, it seems that going across the system one passed by only two or three domains in each direction. Would it not have been useful to study system sizes and or disorder strengths where the localization length $\xi \ll L$?

Overall, the data presented in section 5 is at most qualitative, it shows the one can extract some length which one can define as the localization length, and this length is somewhat correlated with what they get form exact diagonalization. When one tries to compare this with some expectations from localisation theory, such as Eqs. (9) and (10), then again the comparison is only qualitative (and in this case, data is not even shown). Perhaps this is all one could expect to have, but it would be good to have a more detailed discussion of this.

With all the above in mind, I feel that the paper would need to go through a major revision before it can be considered for publication in SciPost. In the requested changes section below I give some more detailed comments and questions.

Requested changes

General requests: 1-Make the paper self-contained. Give details that are essential for understanding what is done in this paper.

2-Provide a motivation and justification for approach taken in Section 4, reinserting $W_E$ into the Schrödinger equation.

3-If possible, provide a discussion of how one could improve the method beyond only nearest neighbor domains in a systematic way.

4-Is it possible to have a more direct comparison with earlier results on localization length?

Further comments/suggestions/questions:

5-On page 2, the Anderson model is said to be "also known as the tight-biding Hamiltonian." It is true that it is "a" tight binding Hamiltonian, but I have never heard it being called "the" tight-binding Hamiltonian.

6-At start of section 2, they describe the system in words and then refer to [52]. The system should be fully described here, and I think it might be useful to write down explicitly the Hamiltonian they use and the form of the disorder potential.

7-In section 3: "Increasing the width of the scatterers $\sigma$ also leads to stronger localisation (not illustrated)." Maybe I'm missing how exactly they are varying $\sigma$ but if $\sigma \rightarrow \infty$ wouldn't they get a constant potential with no localization? So at least at some point one would expect the localization to get weaker with larger $\sigma$?

8-In Fig. 1, it would be useful to add a label on the color bar, and it would also be useful if they plotted the disorder potential for more direct comparison with wave function.

9-In Fig. 2, is the disorder realisation (position of Gaussian potentials) kept fixed while the other parameters are varied? It's not obvious that they are but this would make most sense in the comparison being made.

10-Second paragrap in section 4: "...it appears that $W_E$ may, to a good approximation, be able to replace $V$ in the real Schrödinger equation, directly in the Hamiltonian. " As written above in the main report, I really don't understand this. Why would one want to do this? Why should it be true?

11-In section 4 there is extensive discussion of the "valley lines of $u$" . This is never explicitly defined, but should be.

12-In Figs. 5 and 6, ticks are not visible.

13-In Fig. 7 "candidate paths of least cost" are plotted as green and black lines. Is there any significance to the color?

14- On page 19, the authors mention that "once $E$ exceeds all saddle points, $\xi_E$ diverges to infinity and ceases to exist, at which point our curves must terminate." Since the authors are considering spineless fermions in 2D, all states should strictly speaking be localised in the thermodynamic limit. I guess this means that the above statement would be reflecting finite size effects. It's however not clear to me how one would recover localization here. Can one give some argument for how localization would appear in cases with large energy or will the method fail to detect localized states with energy larger than the amplitude of disorder?

15-I do believe it's important to make comparison with earlier predictions of localization theory where possible, as they authors do with Eqs. (9) and (10). However, no data is provided and only a qualitative discussion. Would it make sense to provide the data?

---

## Round 2 · Referee Report · Anonymous (Referee 2) · 2021-4-7

Report

The authors have significantly updated their manuscript, which is now self-contained and much more clear. The bottom line seems to be that their method works well only for the very few lowest energy eigenstates and fails at higher energies, as expected for the localization landscape. As there is no systematic way of knowing when the method fails, the authors suggest that one needs to visualise the eigenstates, which of course makes the method no longer useful in practice as one would first need to solve the full problem. I guess, though, that the method actually works decently always for the lowest energy eigenstate so one could somewhat rely on that. It is, however, noticeable that the accuracy of their estimates for the localization length, at least as given by the comparison in Table 1, is only within about 10% in many cases.

In Table 1, one comparison entry is missing as the simulation didn't converge. The authors comment that this can be fixed by increasing the accuracy of the simulation. I am surprised that they didn't do so if this is indeed the case.

I still do not understand the logic of putting W_E back into the Schrödinger equation, but I think it's ok to leave the discussion as is.

The paper reports on a method that can be used to obtain qualitative estimates of the localization length for the lowest energy eigenstates in localised systems. The basic idea is based on an earlier work on Agmon distance, with a systematic and efficient way of calculating this Agmon distance in 2D. As such, the manuscript probably satisfies one of the acceptance criteria of SciPost Physics Core, such as "2. Detail one or more new research results significantly advancing current knowledge and understanding of the field." I am not as convinced that it does satisfy any of the acceptance criteria of SciPost Physics. If so, it would likely be "Open a new pathway in an existing or a new research direction, with clear potential for multipronged follow-up work;" but since there are some limitation to their method that doesn't allow for completely systematic and quantitative applications, the potential for multipronged follow-up work is slightly unclear.

---

## Round 2 · Referee Report · Anonymous (Referee 1) · 2021-4-10

Report

In the revised version of the manuscript authors addressed most of the concerns raised in my previous report. In particular they have compared their approximate method for calculating the Agmon distance with the exact result, which shows the accuracy of their method, and clarified the qualitative difference between the true and effective potential, which demonstrates the advantage of using the effective potential for calculating the Agmon distance, as the Agmon distance based on true potential vanishes.

Moreover, authors have performed time-dependent simulations in order to compute the localization length by an established method and compare to the result obtained by their proposed new method. They have found the results to disagree. Because of this authors have limited their claim of applicability of their method to a small range of low energies.

Authors support this claim, by showing that the decay of eigenstates between the domains is in approximate agreement with their LLT-based estimate (Fig. 7), however they point out that the method is prone to failure due to reasons that are extensively discussed in Sec. 6.

This paper introduces several new methods, for instance a way of approximate calculation of the Agmon distance, however, in my opinion, it mostly demonstrates the limitations of the LLT-based approach for calculating the localization length. I don't think it satisfies any of the expectations for publishing in SciPost physics, however, it does satisfy the expectations of SciPost core, and it can be published there if authors address the comments in "requested changes" section of this report.

Requested changes

1 - A comparison between the LLT-based localization length, and the result obtained from the time-dependent simulation should be included, in order to properly illustrate the limitations of the method.

---

## Round 2 · Author Response

We would like to deeply thank both Referees for their insightful questions and comments, which have greatly helped us in improving our paper. Below, we address these methodically.

Referee 1:

The Referee suggests that we should compare the approximate Agmon distance obtained via our LLT-based method to the exact value, arising from solving the semiclassical equations. We have done this for a few examples and the results are shown in Table 1 of the new version of the manuscript. The approximate value is only slightly greater than the true minimal decay cost, and the approximate paths are very close to the exact, classical ones, indicating that our method is performing well.

In the process of answering this question by the Referee, we have spotted an oversight in the previous version of the manuscript, which has now been corrected: the decay coefficient that was used in the LLT calculation of the localisation length was not taken as the smallest integral over all candidate paths arising from LLT, but as an average over these. It was also this average quantity that was plotted in Fig. 8 of the previous version of the manuscript. This was done because the Agmon distance itself underestimates the true decay coefficient, being a formal lower bound. On the other hand, we found that by averaging over all the paths connecting the two domains through the saddle points, we could in fact approximate the real decay coefficient much better. The first author apologises and takes full responsibility for the lapse in memory which led to this inaccuracy in the description given in the previous version in the manuscript. The new version both corrects this issue and comments on it from a physical point of view. We thank the Referee for helping us find and correct this mistake.

Second, the Referee proposes that we should compare our calculations of the localisation length to established methods, pointing out in particular the Transfer Matrix Method (TMM) which is exact and has been used successfully for discrete systems in all dimensions. Making such a comparison is indeed crucial, but we cannot use the TMM because we are not aware of any way to generalise it to 2D continuous potentials. In fact, we do not know of any exact (or at least fairly accurate and reliable) methods that could tackle the problem at hand, except for exact diagonalisation and direct integration of the time-dependent Schrödinger equation.

We have therefore performed time-dependent simulations, initiating a translating Gaussian wavepacket outside the disorder, choosing a large enough system to observe exponential decay, and extracted the associated length scale from the density profiles at long times. This was done in a parameter regime and at energies where the LLT localisation length was smooth and monotonically increasing, not at all affected by the noise seen at high energies (see inset of Fig. 9 in the previous version of the manuscript). We found that the localisation length from time-dependent simulations was considerably greater than our LLT value, by up to as much as an order of magnitude. Being rather surprised by this, we investigated the underlining cause by carefully inspecting the eigenstates, and discovered that our method is limited to much smaller energies than we previously believed. We have adjusted the entire paper to reflect this, as well as added a new section (section 6 in the new version of the manuscript) that described how and why the LLT calculation fails at higher energies. In short, the eigenstates are not governed by pure decay anymore: the quantum tunnelling picture in the effective potential breaks down and the Agmon inequality [equation (7) in the new version of the paper] ceases to be relevant. In this regime, it is best to think of the eigenstate localisation as arising directly from quantum interference effects. Here, we see several mechanisms come into effect that increase the localisation length beyond the prediction based on the clean tunnelling picture. We thank the Referee for asking this question, which helped us correct a serious misconception regarding the applicability regime of the new LLT method.

In terms of providing a comparison of LLT to a different approach at very low energies where the former is applicable, we have performed simulations with very strong noise such that the energy range over which the LLT calculation is correct is sufficiently wide to fit in a slowly moving Gaussian wavepacket. In this case, we have likewise found that time-dependent simulations yielded a larger localisation length than LLT. Having investigated this phenomenon, we realised that when one adds empty “reservoirs” on either side of the noisy region (at x=0 and x=L), the valley network is modified at these edges of the system. In particular, localisation is weaker here compared to the interior of the system due to the modified boundary conditions. When the disorder is very strong and the localisation length is smaller than the typical domain size, these edge effects strongly influence the outcome of the simulation, as the wavefunction only samples these very edges as it rapidly decays exponentially on its way through the noise.

Thus, a meaningful comparison of the LLT method to time-dependent simulations remains elusive (we have included the discussion above in the paper to clarify the situation). However, the comparison to eigenstates at low energies, where the tunnelling picture applies, is not meaningless. Since the domain “diameter” calculation is very transparent at these low energies, and the decay coefficient has been shown to reflect the behaviour in the eigenstates well, we see no reason to question that the ratio of the two indeed gives the localisation length correctly.

Next, the Referee raises concerns over the similarity of V and W_E in the regime where the substitution of the former by the latter works well. This has motivated us to directly compare the two potentials and clarify the connection between them. The effective potential has peak ranges in the same locations where the physical potential has scatterers. The difference is that the effective potential is a “smoothed out” version of the physical one. Whereas V has clear gaps between the scatterers (at reasonable fill factors and scatterer widths), W_E has closed potential ranges encircling domains, which allows for classical trapping in these potential minima. The effective potential also has lower peaks than the physical, as well as a roughly constant background value away from the scatterers, resulting from the smoothing procedure. This insight has already been provided in one of the earlier LLT publications (Ref. [48] in the new version), but we agree that including this information and discussion in the paper is essential and thank the Referee for the idea.

The Referee further suggests that we should try to compute the localisation length (following our LLT prescription) with V in place of W_E, to check whether LLT provides us with any advantage at all. This is a good idea, but because the domains are classically connected in V (due to the gaps between the scatterers) such that the Agmon distance is always zero, even at vanishing energy, a semiclassical tunnelling picture would predict no exponential decay in V. Moreover, one needs the localisation landscape u to find the domains in the first place, as an analogous calculation with 1/V would not give closed domains in the valley network (c.f. gaps between the scatterers). This discussion has been added to the paper.

We have followed the Referees advice of illustrating V, side by side with W_E, and agree that it is constructive to the paper. Many thanks, once again.

Referee 2:

First, we have made the paper self-contained by including very brief descriptions concerning the algorithms used. We agree that it improves the readability considerably and thank the Referee for the suggestion. Please note that we do not plan to publish the long report on arXiv as an original article, because it is far too long for peer review.

Next, the Referee inquires as to the logic behind section 4, in particular pointing out that W_E is also a disordered potential and will induce quantum interference effects of its own. Since W_E is different to V, these could be expected to be different, in which case one wouldn’t expect the substitution to work well. Instead, W_E should be thought of as a classical potential.

We essentially agree with the Referee’s comments, outlined above. However, we believe that the eigenstate comparison is a meaningful exercise, because later in the paper, we apply a semiclassical approximation to tunnelling in the landscape W_E, which is derived by starting from the Schrödinger equation with W_E as the potential. It seems reasonable to us to first ensure that the exact eigenstates are similar before proceeding with the approximation. This has been made clear in the current version of the article. As for the time-evolution in the two potentials, we now know that the tunnelling picture breaks down at quite low energies (see response to Referee 1), well below the energies of the wavepackets used in the previous version of the manuscript. This comparison therefore indeed does not serve any purpose, and has been removed. As a final remark, having realised that beyond some fairly low energy cut-off, the tunnelling picture breaks down and the localisation of these higher eigenstates should be attributed directly to Anderson localisation, it is interesting that the first handful of these beyond-tunnelling eigenstates are actually similar between V and W_E. In other words, quantum interference effects are also similar at low energies, simply due to the fact that W_E bears close resemblance to V, in particular regarding the peak positions. This comment has also been added to the paper. Many thanks to the Referee for the question.

Next, the Referee comments on the approximation of considering tunnelling only between nearest-neighbour domains when computing the localisation length via our new method. We would like to point out that as far as we are aware, the reduction of the problem to adjacent domains does not introduce an extra level of approximation. In the regime where the tunnelling picture applies, such a decomposition will allow to extract the average cost of crossing a single domain wall, and decay over longer distances can then be composed of several such events. In this sense, our calculation is local in nature, and largely independent of system size (more is said about this below). A discussion to this effect has been added to the paper.

The Referee then insightfully comments that he/she expects the method to only work for very low energies and very strongly localised eigenstates. We now know that this is indeed the case, and that at higher energies, the classical trapping in the effective potential picture is no longer helpful. Section 6 in the new version of the manuscript is dedicated to this discussion. We are, however, not aware of any way one could use the Agmon distance to obtain the localisation length in the regime where the mechanisms of section 6 come into effect, primarily because we do not think the eigenstates can be captured by tunnelling in W_E any longer and it is best to think of the physics as quantum interference directly. We greatly thank the Referee for these questions and remarks.

Regarding the new top panel of Fig. 7 (the old Fig. 8), indeed the scatter is strong, but we perform extensive averaging during the calculation of the localisation length, so we do not think this is problematic. The sufficiency of the degree of averaging can be checked by confirming that ξ_E varies smoothly and monotonically with parameters.

We agree that it would have been excellent if we could compare our method to another, established calculation, but as discussed in the reply to Referee 1, to the best of our knowledge, there is nothing else we can do. We have also searched the literature for continuous 2D systems where accurate results for the localisation length are known so that we could apply our method to these as a test, but have not found such examples. Instead, we have added this idea to the future work list, in case an example is uncovered at a later time.

Next, the Referee raises the excellent question of system size compared to the localisation length, and whether the results are reliable if the latter is of the same order or even larger than the former. In principle, as mentioned above, our calculation is local: we only need the system to be large enough to fit in a few domain pairs, such that we can extract several decay coefficients and domain areas. In other words, averaging over many small systems is equivalent to averaging over fewer larger systems. In practice, there are finite size effects that change the structure of the valley network when the system size is comparable to or smaller than the average valley line spacing (studied in Refs. [53,65] in the new version of the manuscript), and we must keep in mind that the system size determines when the noise in the ξ_E(E) curve begins (see Fig. 9 of the previous version of the paper). Importantly, however, we now know that our method stops working long before we reach the system-size-limited noise. Therefore, we have replaced the old Figs. 9 and 10 by the new Fig. 8, which shows only E=0 data, confirmed as meaningful by exact diagonalisation. We have clarified these issues in the article and thank the Referee for the questions.

The Referee points out that the old Fig. 7 (now Fig. 6) shows a systems with only a handful of domains. We chose to use this example so that the details of the network can be clearly seen by eye, but we agree with the Referee that if we increase the strength of the disorder (and thus decrease the localisation length), the validity range of our computation grows, which indeed makes such a regime more useful to study.

As for the comparison of our data with the analytical formula (10) [previously (9)], we have now fitted the LLT data points shown in Fig. 8 and added the curves to the plot. However, we can no longer check the energy dependence of the coefficients in equation (11) [previously (10)] because we now know that our LLT calculation is only valid at very low energies. In addition, we have realised that it is the Boltzmann mean free path which enters equation (10) [previously (9)], and not the scattering mean free path, which means that we would need to compute the scattering cross section of the Gaussian scatterers – not a simple task. Since the formula is very approximate and accurate LLT results are not available above a fairly low-energy cut-off anyway, we decided against developing this further.

As for the remaining (minor) suggestions, points 5-15 on the list of changes in the report, we are grateful to the Referee for bringing these to our attention. We have accepted the Referee’s advice on all points (where possible). The ones that require an answer and have not already been discussed above are addressed presently:

7 – Indeed, increasing either f or σ excessively causes scatterer overlap and a decrease in randomness. A comment to this effect has been added to the paper. 8 – We are deeply sorry, but we genuinely do not understand what label the Referee would like to be added to the colour bar. Instead of plotting the potential V for comparison with the eigenstates (V is now shown in the new Fig. 4 for a different noise realisation), we have added a comment that describes the fact that the weight of the eigenstates lies inside the LLT domains, with the amplitude forced down at the valley lines. Since valley lines correspond to ridges in the effective potential, which is turn occur at the positions of the scatterers in V, we believe this conveys the same information. 9 – No, completely different noise realisations are used, as the fill factor changes between the panels as well. This has been clarified in the article. 12 – Thank you for pointing it out. These figures have now been removed, but we are afraid that there is nothing we can do about this for surface plots plotted in Matlab in general. 13 – The colours used for the valley lines (red and blue) and the candidate paths (green and black) do not carry any distinctive meaning. We simply use different colours to make it easier to see the structure of the network. This has been clarified in the relevant figure captions. 14 – The mobility edge predicted by LLT is indeed not a finite size effect (although it is a natural assumption, one that we ruled out by increasing system size and observing the effect). We have investigated it thoroughly in [53, 65] (numbers correspond to citations in the new version of the manuscript). We have concluded that LLT stops being useful at higher energies. Quantitatively, this happens beyond the narrow energy range where the eigenstates obey pure tunnelling in the effective potential. Qualitatively, as long as there are valley lines remaining, the eigenstates will be pushed down at their positions, so the valley network still provides at least some useful information. However, past the peaks of W_E (note: not V), there is no more eigenstate confinement predicted by LLT, which then loses its value. This does not mean that the eigenstates cannot be suppressed, as LLT theory says that if there are effective valley lines, then the eigenstates are suppressed, but it is not an “if and only if” statement (this was first pointed out to us by Jan Major, whom we thank for his contribution). 15 – We are now showing some limited data in the new version of the paper, but it is inconclusive in terms of supporting or negating the analytical formula (10) [numbering refers to the new version].

We would once again like to deeply thank the Referees for their time, effort and excellent questions and suggestions which we believe have helped to improve our article significantly.

---

## Round 2 · List of Changes

Minor rephrasing (e.g. for better flow) etc. not listed if the content and meaning are not changed.

Title: - Added “at very low energies” to reflect calculation applicability range.

Abstract: - Made the description of section 4 more concrete, leaving only the eigenstate comparison. - Added conditions of applicability of our method. - Added a description of the new section 6, discussing the breakdown of our method at higher energies.

Introduction: - Added the new Ref. [2] in several places. - Added a short paragraph just before the introduction of LLT that highlights the unavailability of other, conventional methods that could accurately capture our system, apart from direct time-integration of the Schrödinger equation. - Clarified in several places that our method is only effective at very low energies. - Modified the description of what is done in section 4, sharpening the motivation and focusing on the eigenstate comparison. [this was done in two places] - Explicitly wrote out the condition on the potential for LLT to apply. - Replaced a reference to the Agmon distance by one to the decay coefficient (the two are not the same). - Added a description of the new section 6 [in two places]. - Removed references to the Report on arXiv, background LLT citations, and external technical appendices, as these are no longer needed.

System of Interest: - Gave a more complete description of the system, including writing out the Hamiltonian.

Exact diagonalisation: - Added a note regarding the unavailability of other useful methods for tackling our problem as motivation for considering exact diagonalisation, with a mention of time-dependent simulations that are mostly left for another paper. - Added a brief note on the algorithm used. - Clarified that in Fig. 2, the noise realisations differ from panel to panel. - Added a discussion of the limit of very large fill factor or scatterer width, in which case the randomness deceases as scatterers overlap, and localisation weakens. - Removed the discussion of possible classical trapping in V, as a better version is now incorporated into the next section (see below). - Figure 2 caption: added a note that the stripes seen in the right-hand-side panels are artefacts of the algorithm. - Last paragraph of section 3: added another argument for why the variance should be treated with caution: the presence of secondary bumps in the eigenstates increases the variance, even if their size and decay rate are identical to the main bump.

The effective potential: - Definition of ‘valley lines of u’ included upon first mention. - Clarified the motivation behind the section. - Added a note on the numerical algorithm used. - Described the connection between V and W_E and discussed the (im)possibility of classical trapping in V, and the fact that WE will also give rise to Anderson localisation on account of being a random potential. - Added a note relating the eigenstates in Fig. 1 to the valley lines, and through these, to the effective and physical potentials. - Added the new Fig. 4 to illustrate V and W_E. - Added some forward references to the new section 6. - Removed the qualitative discussion justifying the existence of the energy shift seen in the new Fig. 5, and replaced with a citation to one of the original LLT papers where it is treated rigorously. - Removed the old Figs. 5 & 6 and all discussion thereof, as wavepackets lie above the energy cut-off where our calculation stops being meaningful. - Added a note to link the better transmission in the effective potential to the lower peaks it has compared to V. - Modified the concluding paragraph to account for changes to the section, and attempted to make the logic of the exercise clearer.

Eigenstate localisation length:

Preamble: - Clarified calculation only works at very low energies. - Removed reference to technical appendix on arXiv. - Added a discussion about the unavailability of other methods to compare our calculation to, and briefly described how far we can get with time-dependent simulations in this regard.

Outline of the LLT method: - Re-emphasised that V will not allow for classical trapping under moderate levels of disorder. - Added a note about the Agmon inequality, to highlight that it is only applicable in situations involving tunnelling, and that previous LLT work has demonstrated its high levels of performance in a 1D example. - Modified the text to make it clear that the Agmon distance need not give the correct decay rate (as an a priori assumption). - Included a comparison of our approximate calculation of the Agmon distance to the (numerically) exact number for a few examples, given in Table 1. - Explained that considering only nearest-neighbour domains does not introduce an extra level of approximation, and that our calculation is local in nature. - Described the test of the Agmon distance as a way of quantifying the decay coefficient between domains (shown in the bottom panel of Fig. 7 in the new version), finding that it underestimates the true value. - Motivated and introduced the “mean” Agmon distance, taken as the average integral over all candidate paths from LLT, and provided a short physical discussion of it. - Pointed out that an analogous calculation starting from V directly (avoiding LLT) is not possible.

Test of decay constants: - Clarified that the calculation is only applicable at low energies for strongly localised states with pure, straight-forward decay only. - Added a second panel to Fig. 7 to check the performance of the Agmon distance proper as an estimate of the decay coefficient, together with a description of this figure. - Added a discussion of how one could use time-dependent simulations to extract the localisation length for comparison to our method, and described the outcome of such a comparison, referring the reader to the new section 6 for an explanation of the observations. - Have corrected throughout instances where \bar{ρ}_E was previously erroneously labelled (and described) as ρ_E.

Effect of parameters: - Removed the old Figs. 9 & 10, and discussion of these, because we now know that our calculation is not meaningful for most of the energy range shown, as is now explained in the text. - Added the new Fig. 8 which focuses on zero-energy results and serves to demonstrate the effect of fill factor and scatterer height, while variation with system size is described in words. - Replaced the extensive discussion of how our calculation becomes inaccurate when limited by system size by a brief description, as we now know that the computation ceases to be relevant at much lower energies. - Energy dependence is no longer illustrated, but merely verbally described. - Changed the way our results are compared to equation (10) [in the new version of the paper]. Since we can no longer do the fits for various values of the energy (the range of applicability is very small and changes with parameters), we only handle the E=0 case and display the best fit curves for it exclusively. Taking into account that we do not know (without performing additional calculations) the scattering cross section of a single Gaussian bump, we have no way of knowing the energy dependence of the Boltzmann mean free path. Moreover, we are not able to extend our calculation to non-zero energy reliably, so we cannot judge whether equation (10) is supported by the numerical data or not. We can only say that reasonable fits are possible at E=0 as a function of fill factor.

Breakdown at higher energies: This entire section is new.

Multidimensional tunnelling: - Added the differential equations in parametric form that need to be solved to obtain the true semiclassical minimal path. - Elaborated on the need to guess the initial direction of the classical trajectory at least twice if the points of interest are separated by one or more turning surfaces. - Added a short discussion of the fact that our “mean” Agmon distance is able to capture the true decay rate, rather than providing a lower bound. - Removed the idea of applying the path decomposition method to our system as we do not think it would be practical.

Conclusions and future work: - Emphasised that our calculation is limited to very low energies. - Added a note on making use of the “mean” Agmon distance and the advantage it provides. - Added a description of the new section 6. - Removed mention of the comparison between time-evolution in V and W_E. - Added the future work idea of perhaps repeating the LLT calculation for a system where the Green’s functions method would be applicable.

---

## Round 3 · Author Response

Editor: Many thanks for pointing out the paper [Phys. Rev. B, 101, 22, 220201(R) (2020)]. We have added a short discussion of it at the end of section 6, as well as a related publication [Phys. Rev. B, 101, 8, 081405 (2020)]. We have also created a Zenodo repository for the code relevant for the work in this paper, which is linked to the resubmitted manuscript.
Referee 1: We thank the Referee for his/her feedback. We would like to comment that while our method does require visualising the eigenstates to determine the range of energies where it is applicable, it is still “useful” in that it computes the localisation length, which cannot be done efficiently from the eigenstates alone.
Regarding the missing entry in Table 1, we thought it was “fair” to acknowledge that such things can happen at the given accuracy, as this is the precision we have used for all presented calculations. However, taking the the Referee’s advice, we have increased the precision and attempted to find the true semiclassical path again. We found that the accuracy had nothing to do with it: the formal minimal path is simply very difficult to find (certainly a pathological example – this does not happen often), and we were again unsuccessful in this task. The reason for the missing entry has been updated in the paper. Many thanks for the inquiry.
Referee 2: We thank the Referee for his/her comments. We have added a quantitative comparison between LLT and time-dependent simulations, as suggested, and agree that it improves the paper.

---

## Round 3 · List of Changes

1) Added a short discussion of [Phys. Rev. B, 101, 22, 220201(R) (2020)] and [Phys. Rev. B, 101, 8, 081405 (2020)] in the last paragraph of section 6, addressing generalisations of LLT, including the ability to handle higher energies. 2) Linked a code repository to the paper upon resubmission. 3) Removed mention of accuracy/precision in the reason for the missing entry in Table 1 given in the caption. 4) Included a description of the system geometry used for time-dependent simulations, a short note on the numerical method employed for this purpose, as well as the functional form of the initial condition at the end of section 2. 5) Added the formula for the energy distribution of the above-mentioned initial condition [the new eqn (11)]. 6) Added a quantitative comparison between LLT and time-dependent simulations for two examples, at the end of sections 5.2 and 6. 7) Administrative change: added a current address for the second author (D.J.B.).

---

## Editorial Decision

published